

# Concentrations of Mg, Ca, Fe, Cu, Zn, P and anthropometric and biochemical parameters in adults with chronic heart failure

Iwona Gorący[1], Ewa Rębacz-Maron[2], Jan Korbecki[3] and Jarosław Gorący[4]

[1] Department of Clinical and Molecular Biochemistry, Pomeranian Medical University, Szczecin, Poland
[2] Institute of Biology, Department of Ecology and Anthropology, University of Szczecin, Szczecin, Poland
[3] Department of Histology and Embryology, Department of Human Morphology and Embryology, Wroclaw Medical University, Wrocław, Poland
[4] Clinic of Cardiology, Pomeranian Medical University, Szczecin, Poland

## ABSTRACT

**Background:** The study investigated the relationship between the concentrations of Mg, Ca, Fe, Cu, Zn, P and anthropometric and biochemical parameters in the blood serum of patients with heart failure (HF) and the potential influence on the development and progression of HF.

**Material & methods:** The study included 214 patients (155 men and 59 women), aged 40–87 years, presenting symptoms or signs typical of HF (according to the NYHA functional classification). Serum concentrations were determined for Mg, Ca, Fe, Cu, Zn, P, C-reactive protein (CRP), creatinine, urea, triglyceride levels (TG), total cholesterol (CH), high density protein (HDL), low density protein (LDL). The levels of macro-and microminerals were analysed using inductively coupled serum optical emission spectrometry (ICP-OES).

**Results:** Our study confirmed the role of known risk factors in the development of heart failure, including: overweight, diabetes, hypertension, high triglycerides (TG), high total cholesterol (CH), high levels of low density protein (LDL) and reduced levels of high density protein (HDL), high CRP, high creatinine. Moreover, deficient serum concentrations of Mg (47% of the studied men and 54% of the women) and Cu (in 44% of men and more than 30% of women) were observed, as well as subnormal serum Fe (2% of women) and Zn (1% of men). Elevated serum Ca was found in 50% of men and 49% of women. In 44% of the studied men and 52% of the studied women, *P* levels in serum were also above-average. The study revealed a significant positive correlation between serum levels of Ca and Mg, and also Ca and Cu in women. In men, serum Cu was positively correlated with Mg and Ca concentrations. In patients from group 1 (NYHA I–II), Mg content was positively correlated with Ca and Cu. In this patient group, Ca was also positively associated with Cu content in serum. In group 2 (NYHA III-IV), serum Mg concentration was significantly positively correlated with that of Cu and Ca.

Corresponding author
Ewa Rębacz-Maron,
ewa.rebacz-maron@usz.edu.pl

**Conclusions:** Changes in the serum concentrations of macro-and microminerals may significantly affect the severity of HF in Polish patients.

## INTRODUCTION

Heart failure (HF) is a disease with a complex etiology and a major public health problem associated with high morbidity and mortality worldwide (*McMurray et al., 2012*; *Ziaeian & Fonarow, 2016*). Over the years, many studies have explored heart failure outcomes in relation to: age, coronary artery disease (CAD), hypertension (HT), valvular heart disease (VD), obesity, diabetes mellitus (DM) and left ventricular mass (LVM) (*Yancy et al., 2013*).

HF is caused by structural or functional cardiac abnormalities which impair the heart's ability to pump and/or fill with blood. During the disease process in HF, the left ventricle (LV) undergoes structural and functional changes involving cardiomyocyte death, fibrosis, inflammation (*Elmadhun et al., 2014*). Pathological left ventricular dysfunction develops in response to various haemodynamic, neurohormonal, genetic and biochemical factors. However, it is known that the neurohormonal factors, including the activation of the renin-angitensin–aldosterone system (RAAS), sympathetic nervous system or antioxidant factors, altered expression of endothelin and inflammatory cytokines, play a fundamental role in LV pathophysiology, leading to its remodelling, hypertrophy and consequently HF (*Guo, Guo & Ji, 2016*). The RAAS is a key factor which contributes not only to blood pressure regulation, renal haemodynamics, fluid or electrolyte homeostasis (regulation of NaCL and water handling) (*Forrester et al., 2018*), but also influences the development of LV remodelling and hypertrophy, stimulating the growth and proliferation of heart cells (*Forrester et al., 2018*). Remodelling and hypertrophy constitute an adaptive response due to myocardial infarction, hypertension, or valvular heart disease. In patients with cardiovascular disease, the RAAS activity is often elevated (*Pugliese, Masi & Taddei, 2020*). Diastolic dysfunction may also be caused by changes in the rate and degree of LV relaxation and passive stiffness (*Verma & Solomon, 2009*; *Lalande & Johnson, 2008*). Ventricle relaxation is related to the active transport of calcium ions to the sarcoplasmic reticulum, which causes dissociation of myosin-actin crossbridges (*Lalande & Johnson, 2008*). Slow or incomplete relaxation caused by changes in calcium homeostasis reduces the atrioventricular pressure gradient in the early diastolic period, which leads to a reduction in LV filling (*Zile, Baicu & Gaasch, 2004*).

In addition, the regulation of HF is under control of different physiological systems. However, the exact pathomechanism of HF development and progression is still intensively studied. So far, we do not have enough data on the influence of minerals on the development and progression of HF. Macro-and microminerals such as magnesium (Mg),

calcium (Ca), iron (Fe), copper (Cu), zinc (Zn), chromium (Cr), manganese (Mn), phosphorus (P) are essential cofactors for energy transfer and physiological heart function, they have antioxidant properties and are involved in multiple signalling pathways. Ca and Mg participate in many physiological processes in the cardiovascular system (*Kolte et al., 2014*; *Reid, Gamble & Bolland, 2016*). Ca plays a central role in the regulation on myocyte contractility, regulation of gene expression, and cellular processes such as growth and apoptosis. However, Ca has also been shown to be involved in regulating the transcription and expression of genes characteristic of the hypertrophic response in cardiomyocytes (*Dahl et al., 2018*). Hypercalcemia is also associated with vascular calcification, potentially increasing the risk of cardiovascular disease (*Seely, 1991*). On the other hand, it has been suggested that Mg promotes vasodilation, partly directly, as a calcium antagonist in smooth muscle cells, and also indirectly, by modulating endothelial function (*Kolte et al., 2014*). It was also reported that high Ca and low Mg levels in serum may increase the risk for coronary disease (*Bolland et al., 2011*; *Del Gobbo et al., 2013*; *Rohrmann et al., 2016*; *Larsson, Burgess & Michaelsson, 2017*; *Larsson, Burgess & Michaëlsson, 2018*), which is the main risk factor for the development of heart failure (*Ziaeian & Fonarow, 2016*). On the other hand, with respect to coronary disease, an inverse relationship was suggested between the dietary intake of Mg and its blood content and the prevalence of heart failure (*Zhang et al., 2012*; *Lutsey et al., 2014*; *Kunutsor, Khan & Laukkanen, 2016*; *Taveira et al., 2016*; *Wannamethee et al., 2018*). However, there are not many studies dedicated to the relationships between these elements and the prevalence of heart failure (*Helte, Åkesson & Larsson, 2019*). In the case of high Ca levels in serum, research points to a trend of an increased risk of HF (*Lutsey et al., 2014*), but findings are not consistent (*Van-Hemelrijck et al., 2013*; *Donneyong et al., 2015*).

Fe is a basic trace element required for the transport, storage and usage of oxygen in humans. In HF, iron deficiency is one of the most common comorbidities, affecting 37–61% of patients (*Jankowska et al., 2010*). The deficiency, even before the onset of anemia, can be severe among patients with HF, aggravating symptoms, adversely affecting quality of life, functional status and clinical outcomes, as well as being associated with an increased risk of mortality (*Peyrin-Biroulet, Williet & Cacoub, 2015*; *Klip et al., 2013*; *Jankowska et al., 2013*). Under the European Society of Cardiology (ESC) Guidelines for the diagnosis and treatment of acute and chronic HF, all patients with HF should be tested for iron deficiency (*Bekfani et al., 2019*).

Cu is an essential trace element, needed for normal heart function. It was demonstrated that copper deficiency is a factor leading to cardiac hypertrophy (*Elsherif et al., 2004*; *Zhou, Bourcy & Kang, 2009*; *Zhou, Bourcy & Kang, 2009*). Cu deficiency also disrupts cardiac calcium metabolism (*Elsherif et al., 2004*) and leads to histopathological changes contributing to HF (*Elsherif et al., 2003*; *He & James Kang, 2013*; *Zhang et al., 2016*). Copper supplementation was shown to improve cardiac function and reduce cardiac hypertrophy (*Hughes et al., 2008*; *He & James Kang, 2013*; *Zheng et al., 2015*).

Zn is an essential element for humans and is required for the activity of many enzymes, affecting multiple signalling pathways and transcription factors, as well as being the second most abundant trace metal in humans (*Cherasse & Urade, 2017*; *Vasto et al., 2006*).

It plays an important role in controlling cardiac contractility in cardiomyocytes. Moreover, studies that included zinc in multiple microelement supplements have suggested an association with improved cardiac function and quality of life (*Jeejeebhoy et al., 2002*).

Patients with HF were reported to exhibit both hypo-and hyperphosphatemia (*Christopoulou et al., 2017*). Hypophosphatemia was observed as a result of post-surgery inflammatory response (*Ess et al., 2013*; *Cohen et al., 2004*; *Polderman & Girbes, 2004*). Underlying diabetes may be another reason for hypophosphatemia, due to the use of sodium-glucose co-transporter two inhibitors in patients with HF (*Weir et al., 2014*), which may impair phosphate uptake from the intestine. It was demonstrated that depleted blood P concentrations interfere with heart function by reducing the amount of phosphorus available for ATP synthesis in cardiomyocytes (*Pesta et al., 2016*). However, 6% of HF patients present with hyperphosphatemia (*Ess et al., 2013*; *Christopoulou et al., 2017*), which may be associated with chronic kidney disease, a frequent co-existing condition in these patients, reducing renal phosphate excretion (*Christopoulou et al., 2017*). In contrast, many studies have shown reduced levels of serum phosphorus in HF patients (*Ess et al., 2013*; *Cohen et al., 2004*; *Polderman & Girbes, 2004*, *Pesta et al., 2016*).

In the present study, we aim to determine the effects of selected elements–Mg, Ca, Fe, Cu, Zn, P, as well as selected biochemical parameters and anthropometric characteristics, on the severity/progression of heart failure in patients with chronic heart failure and the impact of these elements on the risk factors for HF development. It is important to identify links between anthropometric parameters and biochemical parameters in patients, seeing as body size, as well as physiology, biochemistry and patient sex are significant cardiac factors. Our study refers to patients who live and receive treatment in Poland.

Identifying the interplay between anthropometric and biochemical parameters in patients is important in cardiology. Body metrics have a powerful impact on blood biochemistry in both men and women. We have been able to cite extensive literature because the subject analysed in this paper is of great interest to the medical and research community. Our aim, however, was to take a fresh look at data, linking biochemical parameters to anthropometrics for men and women separately according to NYHA classes. We hope other researchers will pick up this subject so that the knowledge on the relationships between patients' laboratory test results and anthropometrics can be gradually expanded.

## MATERIALS AND METHODS

We enrolled prospectively 214 consecutive patients aged 40–87 years (155 men and 59 women) at the Clinic of Cardiac Surgery and Clinic of Cardiology, Pomeranian Medical University in Szczecin, Poland. The inclusion criterion was a diagnosis of chronic heart failure (HF). Our patients were diagnosed by a specialist cardiologist and underwent all the standard and necessary examinations required in the diagnosis of heart failure (including electrocardiography, Doppler echocardiography to evaluate LV dysfunction) and other additional tests depending on patient condition.
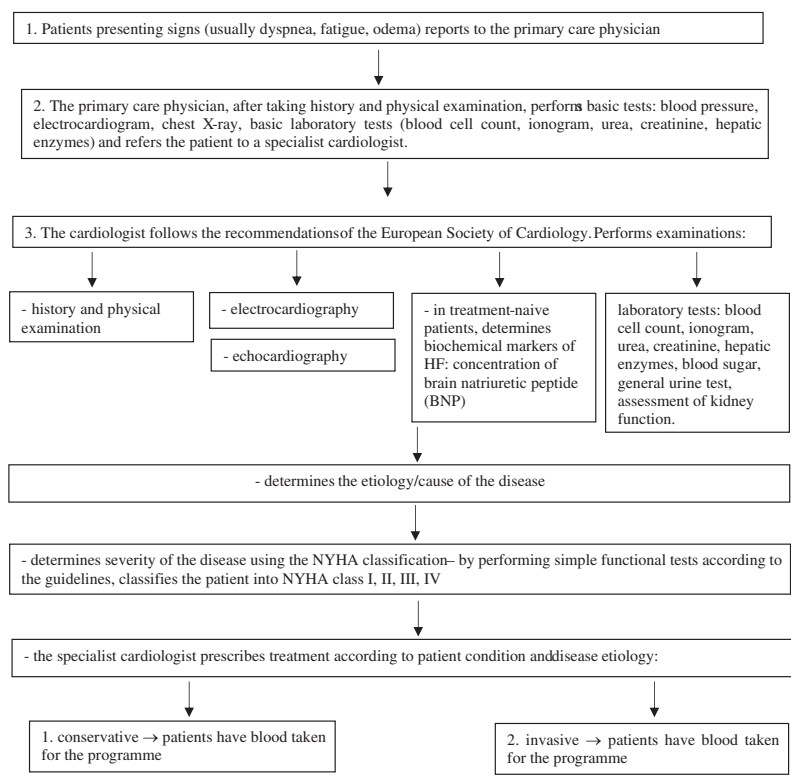

**Figure 1 Heart failure diagnostic procedure according to the guidelines of the European Society of Cardiology (ESC), 2016 ESC Guidelines for the diagnosis and treatment of acute and chronic heart failure.**

## Grouping of study participants

In this program, after a medical examination and additional tests, the doctor administered fitness tests to each patient to determine the NYHA class for each patient. Enrolment and inclusion criteria of study participants are presented in the flow chart (Fig. 1).

In this study, HF patients were divided into two groups: group 1 with patients presenting NYHA class I–II heart failure, and group 2 with patients presenting NYHA class III–IV heart failure. The patients' underlying heart conditions were predominantly: coronary heart disease (CAD) and valvular disease (VD). Among 214 HF patients included in the study, 132 patients had CAD, 58 patients had CAD+VD, and 30 patients had VD. The patients' demographic data and medical history were collected from their medical chart records.

Arterial hypertension was defined as systolic blood pressure exceeding 140 mmHg, and/or diastolic blood pressure greater than 90 mmHg or reported historical data showing hypertension.

The study was based on the diagnostic survey method using questionnaire techniques and laboratory tests (blood samples were collected for biochemical parameters and mineral concentration determinations). In the first step of the procedure, we asked patients to provide some basic data (tobacco smoking, BMI, diabetes, arterial hypertension, history of myocardial infarction). Body Mass Index (BMI) scores, according to WHO, were

interpreted as follows: below 18.5 = underweight; 18.5–24.9 = normal weight; 25.0–29.9 = pre-obesity; 30.0–34.9 = class I obesity; 35.0–39.9 = class II obesity; 40 and above = class III obesity (*WHO, 2020*). Smoking status was categorised as smoker or non-smoker.

Laboratory data determined in all subjects included: C-reactive protein (CRP), creatinine, urea, triglyceride levels (TG), total cholesterol (CH), high density protein (HDL), low density protein (LDL). The following normal values were applied: CH–below 190 mg/dL; TG–below 150 mg/dL; HDL–above 40 mg/dL for men and above 50 mg/dL for women; LDL below 70 mg/dL. Biochemical parameters were determined by a certified laboratory using commercially available standardised methods. Patients with a clinical diagnosis of: cardiomyopathy, coagulopathy, collagenosis and chronic inflammatory disease were excluded from the study.

The study protocol was approved by the Ethics Committee of the Pomeranian Medical University in Szczecin with formal informed consent signed by all participants. The program was approved by the Bioethics Committee of the Pomeranian Medical University by resolution No.KB-0012/87/17 of 19.06.2017.

## Analysis of elemental concentrations in serum

All samples were transferred into 1.5-ml microtubes and stored at −80 °C until processed. Samples were analysed by inductively coupled serum optical emission spectrometry (ICP-OES, ICAP 7,400 Duo; Thermo Scientific, Waltham, MA, USA) using a concentric nebulizer and a cyclonic spray chamber to determine their Mg, Ca, Fe, Cu, Zn, P content. Analysis was performed in radial and axial mode. The samples were thawed at room temperature and digested using a microwave digestion system (MARS 5, CEM, Republic of Chile). The volume of the sample used in the analysis was 0.75 ml. The samples were transferred to clean polypropylene tubes, two mL of 65% $HNO_3$ (Suprapur; Merck, Darmstadt, German) was added to each vial and each sample was allowed 30 min pre-reaction time in the clean hood. At the end of the pre-reaction time, one mL of non-stabilized 30% $H_2O_2$ solution (Suprapur; Merck) was added to each vial. Once all of the reagents had been added, the samples were placed in special Teflon vessels and heated in a microwave digestion system for 35 min at 180 °C (15 min ramp to 180 °C and maintained at 180 °C for 20 min). At the end of digestion, all samples were removed from the digestion oven and allowed to cool to room temperature. In the clean hood, samples were transferred to acid-washed 15 mL polypropylene sample tubes. A further 10-fold dilution was performed prior to ICP-OES measurement. The samples were spiked with an internal standard to provide a final concentration of 0.5 mg/L Yttrium. A total of one ml of 1% Triton (Triton X-100, Sigma, St. Louis, MO, USA) was also added and the samples were diluted to the final volume of 10 mL with 0.075% nitric acid (Suprapur, Merck). Samples were stored in a monitored refrigerator at a nominal temperature of 4 °C until analysis.

Blank samples were prepared by adding concentrated nitric acid to tubes without any sample and subsequent dilution in the same manner as described above.

Multielement calibration standards (ICP multi-element standard solution IV (Merck) and Phosphorus ICP Standard (AccuStandard. Inc. New Haven, CT, USA)) were prepared with different concentrations of inorganic elements in the same manner as blanks and samples.

Deionized water (Direct Q UV, approximately 18.0 MΩ; Millipore) was used in the preparation of all solutions. Samples of reference material (NIST SRM 8,414 Bovine Muscle) were prepared in the same manner as samples. The wavelengths (nm) were Mg 285.213, Ca 396.847, Fe 259.940, Cu 324.754, Zn 213.856, P 178.766.

## Statistical analysis

The measurable variables representing the characteristics of the studied men and women were analysed by calculating arithmetic means (±SD) and $P$-values for the differences between mean values for males and females (Student's $t$-test). For categorical variables, percentage shares were calculated separately for men and women and for groups identified according to NYHA class (HF1 and HF2). The distribution of elemental concentrations data was tested for normality using the Shapiro-Wilk test. The data in the analysed groups and NYHA categories did not follow the normal distribution, and for this reason further analyses were carried out using non-parametric tests: the Kruskal–Wallis ANOVA, Spearman rank correlation test to examine the relationships between the concentrations of the analysed minerals and categorical variables. Student's $t$-test was employed to compare mineral concentrations depending on sex and NYHA class. The statistical significance level was set at $p \leq 0.05$. Statistical analysis of the observations was carried out in Statistica StatSoft 13.1.

## RESULTS

### Analysis of biochemical and anthropometric parameters. Categorical variables

#### Tobacco smoking

In the study group, more than 63% of men and 76% of women declared they did not smoke cigarettes. In groups based on the NYHA classification, non-smoking was reported by more than 68% of the respondents in group 1 (NYHA classes I and II) and 64% in group 2 (NYHA classes III and IV). The observed differences were not statistically significant (Tables 1, 2).

#### Body mass index (BMI)

None of the participants would qualify as underweight according to the adopted BMI categories. Roughly the same number of men and women, percentage-wise, had a normal BMI (respectively: 18.71% and 18.64%). Likewise, very similar percentages of men and women were overweight or obese according to BMI (respectively: 81.29% and 81.36%). More than 84% of the participants in group 1 according to the NYHA classification (NYHA classes I and II), were overweight or obese. In group 2 according to the NYHA classification (NYHA classes III and IV), overweight and obese individuals accounted for 74.29%. The observed differences were not statistically significant (Tables 1–4).

| Table 1 Percentages of the analysed categorical variables divided according to patient sex. | | | |
|---|---|---|---|
| Examined variable | Total ($n$ = 214) % ($n$) | Male ($n$ = 155) | Female ($n$ = 59) |
| Tobacco smoking | | | |
| No-0 | 66.82 (143) | 63.23 (98) | 76.27 (45) |
| Yes-1 | 33.18 (71) | 36.77 (57) | 23.73 (14) |
| BMI (kg/m$^2$) | | | |
| Underweight | – | – | – |
| Normal | 18.69 (40) | 18.71 (29) | 18.64 (11) |
| Overweight | 40.65 (87) | 41.94 (65) | 37.29 (22) |
| Obese | 40.65 (87) | 39.35 (61) | 44.07 (26) |
| Diabetes mellitus (DM) | | | |
| No-0 | 63.08 (135) | 64.52 (100) | 59.32 (35) |
| Yes-1 | 36.91 (79) | 35.48 (55) | 40.68 (24) |
| Hypertension (HT) | | | |
| No-0 | 19.16 (41) | 23.23 (36) | 8.47 (5) |
| Yes-1 | 80.84 (173) | 73.77 (119) | 91.53 (54) |
| Myocardial infarction (MI) | | | |
| No-0 | 39.72 (85) | 41.29 (64) | 35.59 (21) |
| Yes-1 | 60.28 (129) | 58.71 (91) | 64.41 (38) |
| NYHA | | | |
| Group 1 (NYHA I and II) | 67.29 (144) | 70.97 (110) | 57.63 (34) |
| Group 2 (NYHA III and IV) | 32.71 (70) | 29.03 (45) | 42.37 (25) |
| CRP (mg/L) | | | |
| <0.6 Normal-0 | 78.97 (169) | 80.00 (124) | 76.27 (45) |
| Higher than normal-1 | 21.03 (45) | 20.00 (31) | 23.73 (14) |
| Creatinine (mg/dL) | | | |
| <1.4 M and <1.2 F Normal-0 | 92.06 (197) | 90.32 (140) | 96.61 (57) |
| Higher than normal-1 | 7.94 (17) | 9.68 (15) | 3.39 (2) |
| Urea (mg/dL) | | | |
| <40 Normal-0 | 55.14 (118) | 55.48 (86) | 54.24 (32) |
| Higher than normal-1 | 44.86 (96) | 44.52 (69) | 45.76 (27) |
| TG (mg/dL) | | | |
| <150 Normal-0 | 27.57 (59) | 26.45 (41) | 30.51 (18) |
| Higher than normal-1 | 24.77 (53) | 22.58 (35) | 30.51 (18) |
| NA | 47.66 (102) | 50.97 (79) | 38.98 (23) |
| Total cholesterol (mg/dL) | | | |
| <190 Normal-0 | 31.31 (67) | 29.68 (46) | 35.59 (21) |
| Higher than normal-1 | 26.17 (56) | 24.52 (38) | 30.51 (18) |
| NA | 42.52 (91) | 45.80 (71) | 33.90 (20) |
| HDL (mg/dL) | | | |
| >40 for M, >50 for F Normal-0 | 20.56 (44) | 15.48 (24) | 33.90 (20) |
| Higher than normal-1 | 28.51 (61) | 31.61 (49) | 20.34 (12) |
| NA | 50.93 (109) | 52.90 (82) | 45.76 (27) |
| LDL (mg/dL) | | | |
| <70 Normal-0 | 12.15 (26) | 10.97 (17) | 15.25 (9) |
| Abnormal-1 | 38.32 (82) | 37.42 (58) | 40.68 (24) |
| NA | 49.53 (106) | 51.61 (80) | 44.07 (26) |

**Table 2 Percentages of the analysed categorical variables divided according to NYHA classes.**

| Examined variable | Total (n = 214) % (n) | NYHA I and II (n = 114) | NYHA III and IV (n = 70) |
|---|---|---|---|
| Tobacco smoking | | | |
| No-0 | 66.82 (143) | 68.06 (98) | 64.29 (45) |
| Yes-1 | 33.18 (71) | 31.94 (46) | 35.71 (25) |
| BMI (kg/m$^2$) | | | |
| Underweight | – | – | – |
| Normal | 18.69 (40) | 15.28 (22) | 25.71 (18) |
| Overweight | 40.65 (87) | 43.75 (63) | 34.29 (24) |
| Obese | 40.65 (87) | 40.97 (59) | 40.00 (28) |
| Diabetes mellitus (DM) | | | |
| No-0 | 63.08 (135) | 63.20 (91) | 62.86 (44) |
| Yes-1 | 36.91 (79) | 36.80 (53) | 37.14 (26) |
| Hypertension (HT) | | | |
| No-0 | 19.16 (41) | 20.83 (30) | 15.71 (11) |
| Yes-1 | 80.84 (173) | 79.17 (114) | 84.29 (59) |
| Myocardial infarction (MI) | | | |
| No-0 | 39.72 (85) | 57.64 (83) | 65.71 (46) |
| Yes-1 | 60.28 (129) | 42.36 (61) | 34.29 (24) |
| CRP (mg/L) | | | |
| <0.6 Normal-0 | 78.97 (169) | 82.64 (119) | 71.43 (50) |
| Higher than normal-1 | 21.03 (45) | 17.36 (25) | 28.57 (20) |
| Creatinine (mg/dL) | | | |
| <1.4 M and <1.2 F Normal-0 | 92.06 (197) | 92.36 (133) | 91.43 (64) |
| Higher than normal-1 | 7.94 (17) | 7.64 (11) | 8.57 (6) |
| Urea (mg/dL) | | | |
| <40 Normal-0 | 55.14 (118) | 56.25 (81) | 52.86 (37) |
| Higher than normal-1 | 44.86 (96) | 43.75 (63) | 47.14 (33) |
| TG (mg/dL) | | | |
| <150 Normal-0 | 27.57 (59) | 28.47 (41) | 25.71 (18) |
| Higher than normal-1 | 24.77 (53) | 26.39 (38) | 21.43 (15) |
| NA | 47.66 (102) | 45.14 (65) | 52.86 (37) |
| Total cholesterol (mg/dL) | | | |
| <190 Normal-0 | 31.31 (67) | 31.95 (46) | 30.00 (21) |
| Higher than normal-1 | 26.17 (56) | 28.47 (41) | 21.43 (15) |
| NA | 42.52 (91) | 39.58 (57) | 48.57 (34) |
| HDL (mg/dL) | | | |
| >40 for M, >50 for F Normal-0 | 20.56 (44) | 17.36 (25) | 27.14 (19) |
| Higher than normal-1 | 28.51 (61) | 34.03 (49) | 17.14 (12) |
| NA | 50.93 (109) | 48.61 (70) | 55.72 (39) |
| LDL (mg/dL) | | | |
| <70 Normal-0 | 12.15 (26) | 11.81 (17) | 12.86 (9) |
| Abnormal-1 | 38.32 (82) | 42.36 (61) | 30.00 (21) |
| NA | 49.53 (106) | 45.83 (66) | 57.14 (40) |

**Table 3 Arithmetic means of the analysed variables divided according to patient sex.**

| Variable | Total ($n = 214$) | Male ($n = 155$) | | Female ($n = 59$) | | P for the differences between mean values for males and females (Student's t-test) |
|---|---|---|---|---|---|---|
| | mean ± SD | mean ± SD | min.–max. | mean ± SD | min.–max. | |
| Age (years) | 66.21 ± 8.25 | 65.02 ± 8.11 | 40.00–87.00 | 69.32 ± 7.84 | 41.00–84.00 | **0.001** |
| Body weight (kg) | 82.17 ± 15.06 | 85.54 ± 14.67 | 54.00–142.00 | 73.31 ± 12.27 | 44.00–105.00 | **<0.000** |
| Body height (m) | 1.68 ± 0.09 | 1.72 ± 0.06 | 1.57–1.91 | 1.57 ± 0.06 | 1.43–1.70 | **<0.000** |
| BMI (kg/m$^2$) | 29.25 ± 4.54 | 29.04 ± 4.45 | 19.13–45.01 | 29.79 ± 4.78 | 20.76–41.91 | 0.282 |
| RR (systolic) | 114.76 ± 16.30 ($n = 185$) | 114.03 ± 16.26 ($n = 134$) | 37.00–155.00 | 86.44 ± 16.42 ($n = 51$) | 90.00–170.00 | 0.327 |
| RR (diastolic) | 67.00 ± 9.34 ($n = 184$) | 67.23 ± 9.41 ($n = 133$) | 40.00–90.00 | 66.37 ± 9.22 ($n = 51$) | 40.00–80.00 | 0.577 |
| CRP (mg/L) | 5.33 ± 12.46 | 5.28 ± 12.02 | 0.20–81.70 | 5.49 ± 13.64 | 0.20–100.50 | 0.912 |
| Creatinine (mg/dL) | 1.03 ± 0.71 | 1.03 ± 0.57 | 0.65–6.83 | 1.04 ± 0.99 | 0.59–7.56 | 0.866 |
| Urea (mg/dL) | 40.89 ± 16.99 | 40.33 ± 16.68 | 7.50–149.00 | 42.35 ± 17.86 | 12.90–119.00 | 0.438 |
| TG (mg/dL) | 169.55 ± 94.22 ($n = 112$) | 165.98 ± 86.19 ($n = 76$) | 37.00–530.00 | 117.08 ± 110.23 ($n = 36$) | 57.00–665.00 | 0.562 |
| Total cholesterol (mg/dL) | 199.58 ± 61.15 ($n = 123$) | 199.92 ± 60.41 ($n = 84$) | 100.90–400.00 | 198.84 ± 63.51 ($n = 39$) | 100.90–367.00 | 0.925 |
| HDL (mg/dL) | 46.76 ± 16.80 ($n = 105$) | 46.64 ± 17.53 ($n = 73$) | 19.14–149.00 | 47.03 ± 15.26 ($n = 32$) | 20.03–89.00 | 0.914 |
| LDL (mg/dL) | 120.08 ± 57.25 ($n = 108$) | 121.40 ± 56.44 ($n = 75$) | 31.00–255.20 | 117.08 ± 59.83 ($n = 33$) | 48.00–306.00 | 0.720 |

**Note:**
Values in bold indicate statistical significance.

### Diabetes mellitus (DM)

In the study group, 35.48% of men and 40.68% of women had diabetes. In groups 1 and 2 identified according to the NYHA criteria, a similar number of people was affected by diabetes-related problems (respectively: 36.80% and 37.14%). The observed differences were not statistically significant (Tables 1–4).

### Hypertension (HT)

Arterial hypertension was less prevalent among the men from the study group (73.77%) than among the women (91.53%). According to the NYHA classification, more than 79.17% of the participants in group 1 had hypertension, with 84.29% in group 2. The observed differences were not statistically significant (Tables 1–4).

### Myocardial infarction (MI)

Among the studied men, 58.71% reported a history of myocardial infarction, compared to 64.41% of the studied women. According to the adopted NYHA-based criteria, 42.36% in group 1 and 34.29% in group 2 had had a myocardial infarction. The observed differences were not statistically significant (Tables 1–4).

### CRP

The mean CRP in serum samples amounted to 5.28 mg/L (±12.02) for men and 5.49 mg/L (±13.64) for women. In 20% of men and 23.73% of women, CRP levels exceeded the

**Table 4 Arithmetic means of the analysed variables divided according to NYHA classes.**

| Variable | Total (n = 214) | Group 1 (NYHA I and II) (n = 144) | | Group 2 (NYHA III and IV) (n = 70) | | P for the differences between mean values for NYHA 1/2 vs. NYHA 3/4 (Student's t-test) |
|---|---|---|---|---|---|---|
| | mean ± SD | mean ± SD | min.–max. | mean ± SD | min.–max. | |
| Age (years) | 66.21 ± 8.25 | 65.85 ± 8.20 | 40.00–87.00 | 66.94 ± 8.36 | 41.00–84.00 | 0.36 |
| Body weight (kg) | 82.17 ± 15.06 | 83.12 ± 13.67 | 54.00–128.00 | 80.22 ± 17.52 | 44.00–142.00 | 0.19 |
| Body height (m) | 1.68 ± 0.09 | 1.68 ± 0.08 | 1.46–1.90 | 1.66 ± 0.10 | 1.43–1.91 | 0.21 |
| BMI (kg/m$^2$) | 29.25 ± 4.54 | 29.41 ± 4.05 | 19.13–41.32 | 28.91 ± 5.43 | 19.45–45.01 | 0.45 |
| RR (systolic) | 114.76 ± 16.30 (n = 185) | 113.26 ± 15.43 (n = 126) | 37.00–155.00 | 117.95 ± 17.75 (n = 59) | 90.00–170.00 | 0.07 |
| RR (diastolic) | 67.00 ± 9.34 (n = 184) | 67.05 ± 9.80 (n = 126) | 40.00–101.00 | 67.46 ± 9.39 (n = 59) | 40.00–90.00 | 0.78 |
| CRP (mg/L) | 5.33 ± 12.46 | 4.85 ± 13.17 | 0.20–100.50 | 6.32 ± 10.87 | 0.20–75.20 | 0.42 |
| Creatinine (mg/dL) | 1.03 ± 0.71 | 1.02 ± 0.69 | 0.60–7.56 | 1.05 ± 0.76 | 0.59–6.83 | 0.81 |
| Urea (mg/dL) | 40.89 ± 16.99 | 40.58 ± 18.19 | 7.50–149.00 | 41.53 ± 14.32 | 20.00–92.00 | 0.70 |
| TG (mg/dL) | 169.55 ± 94.22 (n = 112) | 175.96 ± 96.69 (n = 79) | 37.00–665.00 | 154.19 ± 87.54 (n = 33) | 49.00–530.00 | 0.27 |
| Total cholesterol (mg/dL) | 199.58 ± 61.15 (n = 123) | 206.41 ± 64.50 (n = 87) | 100.90–400.00 | 183.08 ± 49.16 (n = 36) | 100.90–320.00 | 0.05 |
| HDL (mg/dL) | 46.76 ± 16.80 (n = 105) | 48.02 ± 16.24 (n = 74) | 26.00–149.00 | 43.76 ± 17.97 (n = 31) | 19.14–89.00 | 0.24 |
| LDL (mg/dL) | 120.08 ± 57.25 (n = 108) | 125.35 ± 59.68 (n = 78) | 31.00–306.00 | 106.38 ± 48.67 (n = 30) | 42.00–245.00 | 0.12 |

normal range. In groups defined according to the NYHA classification, 17.36% in group 1 and 28.57% in group 2 had abnormally elevated CRP levels. The observed differences were not statistically significant (Tables 1–4).

### Creatinine

The mean creatinine level in serum samples amounted to 1.03 mg/dL (±0.57) for men, and 1.04 mg/dL (±0.99) for women. Creatinine levels above the normal range were observed in 9.68% of men. Among the women, in turn, as little as 3.39% presented elevated creatinine levels. Elevated creatinine in serum was found in 7.64% of the participants allocated to group 1 based on the NYHA classification, and in 8.57% of those in group 2. The observed differences were not statistically significant (Tables 1–4).

### Urea

The mean serum urea concentration for the studied men amounted to 40.33 mg/dL (±16.68) and for 44.52% of the male participants exceeded the adopted normal range. In the case of the studied women, the mean urea concentration amounted to 42.35 mg/dL (±17.86) and exceeded the normal range in 45.76% of that group. Similar percentages were observed in groups 1 and 2 according to the NYHA classification (respectively: 43.75% and 47.14%). The observed differences were not statistically significant (Tables 1–4).

### Triglycerides (TG)

The mean serum TG concentration for the studied men amounted to 165.98 mg/dL (±86.19) and in 22.58% exceeded the adopted normal range. For the studied women, TG was lower than in the men and amounted to 117.08 mg/dL (±110.23), with elevated levels observed in 30.51% of the women. The observed differences were not statistically significant (Tables 1–4).

### Total cholesterol (Ch)

The mean cholesterol was similar for both the studied men and women, amounting to 199.92 mg/dL (±60.41) and 198.84 mg/dL (±63.51) respectively. In 24.52% of men and 30.51% of women, cholesterol levels exceeded the normal range. The observed differences were not statistically significant (Tables 1–4).

### HDL

The mean serum HDL was similar for men and women, amounting to 46.64 mg/dL (±17.53) for the former and 47.03 mg/dL (±15.26) for the latter. In the case of 68.39% of the men and 79.66% of the women, the observed HDL level was below normal range. The observed differences were not statistically significant (Tables 1–4).

### LDL

The mean serum LDL amounted to 121.40 mg/dL (±56.44) for the studied men, and 117.08 mg/dL (±59.83) for women. In 37.42% of the men and 40.68% of the women, LDL exceeded the normal range. The observed differences were not statistically significant (Tables 1–4).

## Elemental analysis

### Magnesium

The mean serum Mg concentration amounted to 20.09 mg/L (±3.57) for the studied men, and 19.80 mg/L (±3.44) for the women. The observed differences were not statistically significant ($p = 0.59$).

Almost 47% of the studied men and more than 54% of the studied women had serum Mg concentrations below the normal range (20–25 mg/L). In more than 47% of men and more than 40% of women, Mg levels were found to be normal, while 6% of the studied men and 5% of the women had elevated serum Mg (Tables 5–8).

### Calcium

The mean serum Ca amounted to 100.00 mg/L (±18.93) for the studied men and 104.13 mg/L (±27.27) for the women. The observed differences were not statistically significant ($p = 0.21$).

Serum Ca concentrations in 36% of the studied men and 33% of the studied women were below the normal range (89–101 mg/L). In 14% of men and 17% of women, Ca levels were found to be normal, while as many as 50% of the studied men and 49% of the women had higher-than-normal serum Ca. The high Ca levels in the studied men and women affected mean Ca/Mg figures, which amounted to (respectively) 5.02 (±0.76) and 5.36 (±1.52) (Tables 5–8).

**Table 5 Percentages of mineral content categories (lower than normal, normal, higher than normal) for the studied elements divided according to patient sex.**

| Studied element | Total ($n = 214$) % ($n$) | Male ($n = 155$) | Female ($n = 59$) |
|---|---|---|---|
| Mg | | | |
| 1-less than normal | 48.60 (104) | 46.45 (72) | 54.24 (32) |
| 2-normal | 45.79 (98) | 47.74 (74) | 40.68 (24) |
| 3-higher than normal | 5.61 (12) | 5.81 (9) | 5.08 (3) |
| Ca | | | |
| 1-less than normal | 35.05 (75) | 35.48 (55) | 33.90 (20) |
| 2-normal | 14.95 (32) | 14.19 (22) | 16.95 (10) |
| 3-higher than normal | 50.00 (107) | 50.32 (78) | 49.15 (29) |
| Fe | | | |
| 1-less than normal | 0.47 (1) | – | 1.69 (1) |
| 2-normal | 28.04 (60) | 24.52 (38) | 37.29 (22) |
| 3-higher than normal | 71.50 (153) | 75.48 (117) | 61.02 (36) |
| Cu | | | |
| 1-less than normal | 40.19 (86) | 43.87 (68) | 30.51 (18) |
| 2-normal | 52.34 (112) | 49.03 (76) | 61.02 (36) |
| 3-higher than normal | 7.48 (16) | 7.10 (11) | 8.47 (5) |
| Zn | | | |
| 1-less than normal | 0.47 (1) | 0.65 (1) | – |
| 2-normal | 20.09 (43) | 21.29 (33) | 16.95 (10) |
| 3-higher than normal | 79.44 (170) | 78.06 (121) | 83.05 (49) |

## Iron

The mean serum Fe amounted to 2.35 mg/L ($\pm1.34$) for the studied men and 2.19 mg/L ($\pm1.17$) for the women. The observed differences were not statistically significant ($p = 0.42$).

Only a single woman had a subnormal Fe level, and no such deficits were observed in men. More than 24% of men and 37% of women had Fe concentrations in the normal range (0.35–1.5 mg/L). For 75% of men and 61% of women, the determined Fe levels were above the normal range (Tables 5–8).

## Copper

The mean serum Cu content for the studied men amounted to 1.31 ($\pm5.29$) and was lower than the mean level in women 2.34 mg/L ($\pm10.05$). The observed differences were not statistically significant ($p = 0.33$).

For 44% of men and 31% of women, the determined Cu levels were below the normal range. Just under 49% of the studied men and the majority of women (61%) had normal copper levels. In a small number of the men (7%) and women (8%), serum Cu concentrations were above the normal range (Tables 5–8).

**Table 6 Percentages of mineral content categories (lower than normal, normal, higher than normal) for the studied elements divided according to NYHA classes.**

| Studied element | Total (n = 214) % (n) | NYHA I and II (n = 144) | NYHA III and IV (n = 70) |
|---|---|---|---|
| **Mg** | | | |
| 1-less than normal | 48.60 (104) | 47.92 (69) | 50.0 (35) |
| 2-normal | 45.79 (98) | 44.44 (64) | 48.57 (34) |
| 3-higher than normal | 5.61 (12) | 7.64 (11) | 1.43 (1) |
| **Ca** | | | |
| 1-less than normal | 34.05 (75) | 35.42 (51) | 34.29 (24) |
| 2-normal | 14.95 (32) | 13.89 (20) | 17.14 (12) |
| 3-higher than normal | 50.00 (107) | 50.69 (73) | 48.57 (34) |
| **Fe** | | | |
| 1-less than normal | 0.47 (1) | – | 1.43 (1) |
| 2-normal | 28.04 (60) | 25.69 (37) | 32.86 (23) |
| 3-higher than normal | 71.50 (153) | 74.31 (107) | 65.71 (46) |
| **Cu** | | | |
| 1-less than normal | 40.19 (86) | 40.97 (59) | 38.57 (27) |
| 2-normal | 52.34 (112) | 51.39 (74) | 54.29 (38) |
| 3-higher than normal | 7.48 (16) | 7.64(11) | 7.14 (5) |
| **Zn** | | | |
| 1-less than normal | 0.47 (1) | – | 1.43 (1) |
| 2-normal | 20.09 (43) | 21.53 (31) | 17.14 (12) |
| 3-higher than normal | 49.44 (170) | 78.47 (113) | 81.43 (57) |

**Table 7 Mean concentrations of the analysed elements according to patient sex.**

| Element (mg/L) | Total (n = 214) mean ± SD | Male (n = 155) mean ± SD | min.–max. | Female (n = 59) mean ± SD | min.–max. | P for the differences between mean values for males and females (Student's t-test) |
|---|---|---|---|---|---|---|
| Mg | 20.01 ± 3.53 | 20.09 ± 3.57 | 7.15–33.49 | 19.80 ± 3.44 | 9.06–29.51 | 0.59 |
| Ca | 101.14 ± 21.57 | 100.0 ± 18.93 | 28.78–162.69 | 104.13 ± 27.27 | 73.28–257.38 | 0.21 |
| Ca/Mg | 5.11 ± 1.03 | 5.02 ± 0.76 | 3.83–8.55 | 5.36 ± 1.52 | 3.80–13.13 | **0.03** |
| Fe | 2.31 ± 1.30 | 2.35 ± 1.34 | 0.51–9.49 | 2.19 ± 1.17 | 0.74–5.96 | 0.42 |
| Cu | 1.59 ± 6.92 | 1.31 ± 5.29 | 0.05–66.50 | 2.34 ± 10.05 | 0.02–78.08 | 0.33 |
| Zn | 1.96 ± 1.59 | 1.91 ± 1.57 | 0.62–10.70 | 2.07 ± 1.65 | 0.98–9.64 | 0.51 |
| P | 194.10 ± 58.40 | 191.57 ± 59.36 | 64.69–469.95 | 200.75 ± 55.74 | 83.31–339.01 | 0.31 |

**Note:**
Values in bold indicate statistical significance.

### Zinc

The mean serum Zn content for the studied men amounted to 1.91 mg/L (±1.57) and was marginally lower than the mean serum Zn level found in women 2.07 mg/L (±1.65). The observed differences were not statistically significant ($p = 0.51$).

**Table 8 Mean concentrations of the analysed elements according to NYHA classes.**

| Element (mg/L) | Total (n = 214) | NYHA I and II (n = 144) | | NYHA III and IV (n = 70) | | P for the differences between mean values for NYHA 1/2 vs. NYHA 3/4 (Student's t-test) |
|---|---|---|---|---|---|---|
| | mean±SD | mean±SD | min.–max. | mean±SD | min.–max. | |
| Mg | 20.01 ± 3.53 | 20.25 ± 3.75 | 7.15–33.49 | 19.52 ± 2.99 | 9.06–26.20 | 0.16 |
| Ca | 101.14 ± 21.57 | 100.99 ± 19.70 | 28.78–167.08 | 101.44 ± 25.13 | 65.68–257.38 | 0.89 |
| Ca/Mg | 5.11 ± 1.03 | 5.04 ± 0.85 | 3.80–8.84 | 5.26 ± 1.32 | 3.98–13.13 | 0.14 |
| Fe | 2.31 ± 1.30 | 2.33 ± 1.26 | 0.61–8.70 | 2.25 ± 1.39 | 0.51–9.49 | 0.67 |
| Cu | 1.59 ± 6.92 | 1.37 ± 5.49 | 0.02–66.50 | 2.05 ± 9.24 | 0.05–78.08 | 0.50 |
| Zn | 1.96 ± 1.59 | 1.93 ± 1.66 | 0.74–10.70 | 2.01 ± 1.44 | 0.62–9.64 | 0.73 |
| P | 194.10 ± 58.40 | 194.30 ± 60.54 | 64.69–469.95 | 193.68 ± 54.13 | 83.55–306.47 | 0.94 |

Only a single man had a Zn level below the reference range (0.66–1.1 mg/L), and none of the studied women were found to be zinc-deficient.More than 21% of men and 17% of women had Zn concentrations in the normal range. For 78% of men and 83% of women, the determined Zn levels were above the normal range (Tables 5–8).

### Phosphorus

In both groups, men and women, the recorded mean P concentrations were high, respectively: 191.57 mg/L (±59.36) and 200.75 mg/L (±55.74). The observed differences in serum $P$ levels in men and women were not statistically significant ($p = 0.31$). Among the studied women, 51% had P concentrations above the mean, with 28% above the third quartile. In men, it was respectively 44% and 24%.

### Correlation analysis

There are different interpretations of the degree of correlation intensity in the literature. In the manuscript, we used the following correlations intervals: $\rho = 0$ no correlation; $0 < \rho < 0.3$ week correlation; $0.3 \leq \rho < 0.5$ moderate correlation; $0.5 \leq \rho < 0.7$ strong correlation; $0.7 \leq \rho < 0.9$ very strong correlation (Stanisz, 2006).

In the studied men, correlation analysis revealed a statistically significant weak and moderate relationship for BMI *vs.* smoking, diabetes, hypertension, TG, HDL (respectively: $\rho = -0.19$, $p = 0.02$; $\rho = 0.27$, $p = 0.00$; $\rho = 0.23$, $p = 0.00$ and $\rho = 0.25$, $p = 0.03$ $\rho = -0.42$, $p = 0.00$). There was also a statistically significant moderate positive correlation between TG and CH, LDL (respectively: $\rho = 0.44$, $p = 0.00$; $\rho = 0.41$, $p = 0.00$) as well as very strong relationship between CH and LDL ($\rho = 0.75$, $p = 0.00$). In the analysis, we found significant weak correlations for hypertension *vs.* CH ($\rho = -0.23$, $p = 0.04$), *vs.* Ca ($\rho = 0.20$, $p = 0.01$) *vs.* Ca/Mg ($\rho = 0.21$, $p = 0.01$). Weak correlations were revealed for myocardial infarction *vs.* HDL ($\rho = -0.25$; $p = 0.03$). Furthermore, our analysis identified a moderate relationship for creatinine *vs.* urea ($\rho = 0.53$, $p = 0.00$) and weak relationships *vs.* LDL ($\rho = -0.23$, $p = 0.04$) and *vs.* P ($\rho = -0.16$, $p = 0.04$). In turn, NYHA was negatively correlated with Fe and positively correlated with Zn and CRP (respectively: $\rho = -0.16$, $p = 0.05$; $\rho = 0.16$, $p = 0.04$; $\rho = 0.29$, $p = 0.00$). Strong positive relationships were observed for Cu *vs.* Mg ($\rho = 0.62$, $p = 0.00$) and *vs.* Ca ($\rho = 0.62$, $p = 0.00$) and also for Ca *vs.* Mg ($\rho = 0.68$, $p = 0.00$). Moderate and weak (respectively) positive

correlations were recorded for Mg *vs.* Fe (ρ = 0.40, *p* = 0.00), Fe *vs.* Cu (ρ = 0.18, *p* = 0.03) Ca *vs.* Ca/Mg (ρ = 0.46, *p* = 0.00), Ca *vs.* Fe (ρ = 0.45, *p* = 0.00). Weak negative correlations were found for Mg *vs.* P (ρ = −0.16, *p* = 0.05) and Mg *vs.* Ca/Mg (ρ = −0.25, *p* = 0.00) (Table 9).

Correlation analysis of the data observed in women revealed a weak positive correlation between BMI and serum CRP (ρ = 0.27, *p* = 0.04) and a strong correlation between BMI and serum TG (ρ = 0.55, *p* = 0.00). Tobacco smoking in women was moderate negatively correlated with serum creatinine in women (ρ = −0.38, *p* = 0.00) and weakly correlated with serum Ca (ρ = 0.26, *p* = 0.05). Diabetes in women also showed a weak and moderate (respectively) positive correlation with creatinine, urea and Ca/Mg ratio in serum (respectively: ρ = 0.33, *p* = 0.01; ρ = 0.42, *p* = 0.00 and ρ = 0.37, *p* = 0.00). Hypertension in women was weakly correlated with urea concentration in serum (ρ=0.28, *p* = 0.03). A very strong positive correlation was observed for CH *vs.* LDL (ρ = 0.75, *p* = 0.00). A significant relationship was also found for HDL *vs.* Ca, Fe and Cu (respectively: ρ = −0.41, *p* = 0.02; ρ = −0.36, *p* = 0.04; ρ = −0.46, *p* = 0.01). Weak negative correlations were also uncovered for CRP *vs.* Fe concentration and Ca/Mg ratio (respectively: ρ = −0.30, *p* = 0.02; ρ = −0.33, *p* = 0.01). Strong relationships were identified in women between Mg and Ca (ρ = 0.60, *p* = 0.00) and also between Mg and Cu (ρ = 0.58, *p* = 0.00). In turn, Ca was correlated with Fe, Cu, Zn (respectively: ρ = 0.43, *p* = 0.00; ρ = 0.65, *p* = 0.00; ρ = 0.37, *p* = 0.00). Fe concentration was weakly and moderately (respectively) correlated with Cu, Zn and P (respectively: ρ = 0.32, *p* = 0.02; ρ=0.32, *p* = 0.02; ρ = −0.39, *p* = 0.00) (Table 10).

Correlation analysis carried out for the variables in group 1 according to the NYHA classification revealed a statistically significant weak and moderate (respectively) relationship for BMI *vs.* diabetes, hypertension, TG, HDL, Ca and Ca/Mg (respectively: ρ = 0.29, *p* = 0.00; ρ = 0.29, *p* = 0.00; ρ = 0.32, *p* = 0.00; ρ = −0.46, *p* = 0.00; ρ = 0.19, *p* = 0.02; ρ = 0.17, *p* = 0.04). In the same group, smoking was weakly and moderately negatively (respectively) correlated with diabetes, creatinine, urea (respectively: ρ = −0.18, *p* = 0.03; ρ = −0.23, *p* = 0.00; ρ = −0.36, *p* = 0.00) and also with Cu (ρ = 0.21, *p* = 0.01). For diabetes, positive correlations were noted *vs.* hypertension, creatinine, urea and Ca/Mg (respectively: ρ = 0.21, *p* = 0.01; ρ = 0.28, *p* = 0.00; ρ = 0.21, *p* = 0.01 and ρ = 0.22, *p* = 0.01). Hypertension showed weak positive correlations with Ca and Ca/Mg (respectively: ρ = 0.24, *p* = 0.00; ρ = 0.19, *p* = 0.02). Looking at biochemical parameters, special attention should be brought to significant moderate correlations for TG *vs.* CH and TG *vs.* LDL (respectively: ρ = 0.54, *p* = 0.00 and ρ = 0.46, *p* = 0.00) and a very strong correlation between CH and LDL (ρ = 0.81, *p* = 0.00). We also identified a significant negative relationship for HDL *vs.* Ca and Ca/Mg (respectively: ρ = −0.30, *p* = 0.01; ρ = −0.24, *p* = 0.04). CRP was positively correlated with Cu (ρ = 0.28, *p* = 0.00) and negatively with Ca/Mg and Fe (respectively: ρ = −0.18, *p* = 0.03; ρ = −0.21, *p* = 0.01). Creatinine showed a weak negative correlation with P (ρ = −0.20, *p* = 0.02) and a moderate correlation with urea (ρ = 0.53, *p* = 0.00). In terms of the analysed minerals in this group, strong and statistically significant relationships were observed for Mg *vs.* Ca and Cu (respectively: ρ = 0.68, *p* = 0.00; ρ = 0.67, *p* = 0.00) with a moderate relationship

**Table 9 Correlation analysis of the studied parameters in men.**

| | Smoking | Diabetes mellitus | Hyper-tension | MI | NYHA | CRP | Creatinine | Urea | TG | CH | HDL | LDL | Mg | Ca | Ca/Mg | Fe | Cu | Zn | P |
|---|---|---|---|---|---|---|---|---|---|---|---|---|---|---|---|---|---|---|---|
| BMI | **−0.19** p = 0.02 | **0.27** p = 0.00 | **0.23** p = 0.00 | 0.15 p = 0.07 | −0.07 p = 0.36 | −0.01 p = 0.89 | 0.06 p = 0.43 | 0.02 p = 0.80 | **0.25** p = 0.03 | −0.01 p = 0.95 | **−0.42** p = 0.00 | 0.03 p = 0.81 | −0.04 p = 0.62 | 0.01 p = 0.91 | 0.05 p = 0.53 | −0.10 p = 0.24 | −0.04 p = 0.63 | 0.02 p = 0.76 | 0.07 p = 0.42 |
| Smoking | | **−0.17** p = 0.03 | −0.06 p = 0.49 | 0.04 p = 0.62 | 0.07 p = 0.37 | 0.14 p = 0.09 | **−0.17** p = 0.04 | **−0.25** p = 0.00 | −0.16 p = 0.16 | −0.10 p = 0.38 | **0.32** p = 0.01 | −0.14 p = 0.22 | 0.15 p = 0.06 | 0.15 p = 0.07 | 0.06 p = 0.49 | 0.08 p = 0.31 | **0.18** p = 0.02 | −0.04 p = 0.65 | 0.07 p = 0.41 |
| Diabetes mellitus | | | **0.25** p = 0.00 | 0.12 p = 0.15 | 0.01 p = 0.85 | **−0.19** p = 0.02 | 0.15 p = 0.06 | 0.07 p = 0.38 | 0.02 p = 0.90 | −0.04 p = 0.70 | **−0.25** p = 0.03 | 0.05 p = 0.66 | −0.10 p = 0.21 | 0.04 p = 0.59 | **0.17** p = 0.03 | −0.06 p = 0.49 | −0.07 p = 0.38 | 0.10 p = 0.22 | 0.07 p = 0.37 |
| Hypertension | | | | 0.09 p = 0.27 | −0.04 p = 0.61 | −0.13 p = 0.10 | 0.09 p = 0.25 | 0.14 p = 0.09 | −0.13 p = 0.27 | **−0.23** p = 0.04 | −0.21 p = 0.07 | −0.21 p = 0.07 | 0.07 p = 0.41 | **0.20** p = 0.01 | **0.21** p = 0.01 | 0.11 p = 0.19 | 0.07 p = 0.39 | 0.05 p = 0.52 | 0.02 p = 0.76 |
| MI | | | | | −0.06 p = 0.43 | −0.05 p = 0.57 | 0.06 p = 0.43 | −0.05 p = 0.57 | 0.11 p = 0.34 | 0.02 p = 0.84 | **−0.25** p = 0.03 | −0.04 p = 0.73 | 0.01 p = 0.91 | 0.13 p = 0.11 | 0.12 p = 0.12 | −0.02 p = 0.79 | −0.02 p = 0.79 | 0.09 p = 0.25 | −0.09 p = 0.28 |
| NYHA | | | | | | 0.29 p = 0.00 | −0.06 p = 0.42 | 0.09 p = 0.24 | −0.03 p = 0.80 | 0.01 p = 0.92 | −0.08 p = 0.52 | −0.02 p = 0.89 | −0.08 p = 0.32 | −0.09 p = 0.24 | 0.01 p = 0.93 | **−0.16** p = 0.05 | 0.05 p = 0.55 | **0.16** p = 0.04 | 0.07 p = 0.36 |
| CRP | | | | | | | 0.07 p = 0.38 | **0.17** p = 0.03 | 0.06 p = 0.59 | −0.13 p = 0.25 | −0.02 p = 0.89 | −0.17 p = 0.15 | 0.01 p = 0.87 | −0.05 p = 0.52 | −0.05 p = 0.52 | **−0.17** p = 0.03 | **0.29** p = 0.00 | −0.01 p = 0.91 | −0.04 p = 0.59 |
| Creatinine | | | | | | | | **0.53** p = 0.00 | −0.03 p = 0.81 | −0.07 p = 0.50 | −0.01 p = 0.96 | **−0.23** p = 0.04 | −0.01 p = 0.89 | 0.04 p = 0.61 | 0.07 p = 0.39 | −0.00 p = 0.98 | 0.09 p = 0.26 | −0.11 p = 0.16 | **−0.16** p = 0.04 |
| Urea | | | | | | | | | −0.18 p = 0.12 | −0.12 p = 0.26 | −0.03 p = 0.78 | −0.15 p = 0.19 | −0.05 p = 0.54 | −0.10 p = 0.22 | −0.04 p = 0.61 | −0.06 p = 0.45 | 0.04 p = 0.62 | −0.03 p = 0.75 | −0.02 p = 0.77 |
| TG | | | | | | | | | | **0.44** p = 0.00 | −0.20 p = 0.10 | **0.41** p = 0.00 | 0.01 p = 0.94 | 0.02 p = 0.84 | −0.00 p = 0.98 | 0.08 p = 0.49 | 0.02 p = 0.84 | 0.07 p = 0.55 | −0.04 p = 0.71 |
| CH | | | | | | | | | | | 0.09 p = 0.46 | **0.75** p = 0.00 | 0.17 p = 0.12 | 0.02 p = 0.88 | −0.04 p = 0.75 | −0.09 p = 0.49 | 0.20 p = 0.07 | −0.15 p = 0.17 | 0.11 p = 0.32 |
| HDL | | | | | | | | | | | | 0.07 p = 0.56 | −0.06 p = 0.62 | −0.15 p = 0.22 | −0.22 p = 0.06 | −0.11 p = 0.37 | −0.09 p = 0.45 | −0.18 p = 0.12 | −0.03 p = 0.80 |
| LDL | | | | | | | | | | | | | 0.11 p = 0.35 | 0.02 p = 0.86 | −0.02 p = 0.86 | 0.03 p = 0.80 | 0.05 p = 0.69 | −0.12 p = 0.29 | 0.15 p = 0.21 |
| Mg | | | | | | | | | | | | | | **0.68** p = 0.00 | **−0.25** p = 0.00 | **0.40** p = 0.00 | **0.62** p = 0.00 | 0.04 p = 0.64 | **−0.16** p = 0.05 |
| Ca | | | | | | | | | | | | | | | **0.46** p = 0.00 | **0.45** p = 0.00 | **0.62** p = 0.00 | **0.16** p = 0.05 | −0.15 p = 0.06 |
| Ca/Mg | | | | | | | | | | | | | | | | 0.12 p = 0.14 | 0.14 p = 0.09 | 0.13 p = 0.10 | 0.03 p = 0.71 |
| Fe | | | | | | | | | | | | | | | | | **0.18** p = 0.03 | 0.06 p = 0.42 | **−0.26** p = 0.00 |
| Cu | | | | | | | | | | | | | | | | | | **0.16** p = 0.03 | −0.11 p = 0.17 |
| Zn | | | | | | | | | | | | | | | | | | | 0.11 p = 0.18 |
| P | | | | | | | | | | | | | | | | | | | |

**Note:**
Values in bold indicate statistical significance.

**Table 10 Correlation analysis of the studied parameters in women.**

| | Smoking | Diabetes mellitus | Hypertension | MI | NYHA | CRP | Creatinine | Urea | TG | CH | HDL | LDL | Mg | Ca | Ca/Mg | Fe | Cu | Zn | P |
|---|---|---|---|---|---|---|---|---|---|---|---|---|---|---|---|---|---|---|---|
| BMI | -0.18 p=0.18 | 0.24 p=0.07 | -0.03 p=0.81 | 0.05 p=0.69 | -0.21 p=0.12 | **0.27** p=0.04 | -0.00 p=0.98 | -0.07 p=0.58 | **0.55** p=0.00 | 0.15 p=0.35 | -0.05 p=0.79 | 0.13 p=0.48 | 0.14 p=0.28 | 0.20 p=0.12 | 0.18 p=0.17 | -0.04 p=0.77 | 0.12 p=0.36 | 0.05 p=0.70 | 0.12 p=0.38 |
| Smoking | | -0.14 p=0.30 | 0.03 p=0.84 | -0.17 p=0.21 | 0.08 p=0.55 | 0.04 p=0.74 | **-0.38** p=0.00 | -0.22 p=0.09 | 0.04 p=0.84 | 0.04 p=0.79 | -0.26 p=0.15 | -0.12 p=0.52 | 0.07 p=0.62 | **0.26** p=0.05 | 0.15 p=0.26 | 0.09 p=0.48 | 0.20 p=0.13 | 0.09 p=0.50 | 0.03 p=0.81 |
| Diabetes mellitus | | | 0.25 p=0.05 | 0.18 p=0.18 | 0.10 p=0.47 | -0.04 p=0.75 | **0.33** p=0.01 | **0.42** p=0.00 | 0.09 p=0.62 | -0.20 p=0.21 | 0.14 p=0.45 | -0.30 p=0.09 | -0.10 p=0.47 | 0.12 p=0.36 | **0.37** p=0.00 | 0.06 p=0.67 | -0.03 p=0.82 | -0.20 p=0.14 | -0.09 p=0.49 |
| Hypertension | | | | -0.16 p=0.24 | 0.23 p=0.08 | 0.18 p=0.18 | 0.22 p=0.09 | **0.28** p=0.03 | -0.14 p=0.42 | -0.07 p=0.66 | 0.03 p=0.88 | -0.13 p=0.48 | 0.01 p=0.94 | 0.08 p=0.57 | 0.06 p=0.67 | 0.08 p=0.54 | 0.17 p=0.19 | 0.06 p=0.63 | -0.13 p=0.32 |
| MI | | | | | **-0.29** p=0.03 | -0.01 p=0.94 | 0.08 p=0.56 | 0.19 p=0.16 | 0.23 p=0.17 | 0.04 p=0.82 | 0.21 p=0.25 | -0.02 p=0.93 | 0.11 p=0.42 | -0.05 p=0.71 | -0.10 p=0.43 | 0.01 p=0.97 | -0.14 p=0.31 | -0.09 p=0.51 | -0.00 p=0.98 |
| NYHA | | | | | | 0.07 p=0.62 | 0.04 p=0.77 | -0.16 p=0.22 | -0.24 p=0.16 | **-0.48** p=0.00 | -0.17 p=0.35 | -0.34 p=0.05 | -0.08 p=0.55 | 0.04 p=0.76 | 0.13 p=0.34 | -0.00 p=0.97 | 0.05 p=0.73 | 0.17 p=0.21 | 0.13 p=0.33 |
| CRP | | | | | | | 0.07 p=0.77 | 0.12 p=0.38 | 0.18 p=0.30 | 0.05 p=0.76 | -0.12 p=0.52 | 0.18 p=0.30 | -0.04 p=0.78 | -0.26 p=0.05 | **-0.33** p=0.01 | **-0.30** p=0.02 | 0.10 p=0.45 | -0.02 p=0.90 | 0.06 p=0.67 |
| Creatinine | | | | | | | | **0.65** p=0.00 | -0.11 p=0.51 | -0.00 p=0.98 | -0.11 p=0.53 | 0.30 p=0.09 | 0.03 p=0.82 | 0.00 p=0.97 | 0.04 p=0.76 | 0.06 p=0.64 | 0.08 p=0.52 | 0.03 p=0.83 | -0.21 p=0.11 |
| Urea | | | | | | | | | -0.21 p=0.23 | -0.03 p=0.87 | 0.02 p=0.92 | 0.13 p=0.47 | -0.00 p=0.99 | -0.11 p=0.40 | -0.03 p=0.81 | 0.09 p=0.51 | 0.03 p=0.82 | -0.23 p=0.08 | **-0.30** p=0.02 |
| TG | | | | | | | | | | **0.39** p=0.02 | -0.16 p=0.37 | 0.17 p=0.35 | 0.18 p=0.29 | 0.31 p=0.07 | 0.18 p=0.30 | 0.00 p=0.99 | 0.16 p=0.36 | -0.02 p=0.89 | -0.04 p=0.83 |
| CH | | | | | | | | | | | 0.13 p=0.47 | **0.75** p=0.00 | 0.03 p=0.86 | 0.22 p=0.17 | 0.15 p=0.36 | -0.06 p=0.73 | 0.04 p=0.82 | -0.22 p=0.17 | 0.15 p=0.35 |
| HDL | | | | | | | | | | | | 0.06 p=0.74 | -0.28 p=0.12 | **-0.41** p=0.02 | -0.19 p=0.29 | **-0.36** p=0.04 | **-0.46** p=0.01 | -0.33 p=0.06 | 0.16 p=0.39 |
| LDL | | | | | | | | | | | | | 0.31 p=0.08 | 0.24 p=0.19 | -0.08 p=0.67 | -0.06 p=0.76 | 0.12 p=0.49 | 0.01 p=0.94 | -0.02 p=0.91 |
| Mg | | | | | | | | | | | | | | **0.60** p=0.00 | **-0.29** p=0.03 | 0.25 p=0.06 | **0.58** p=0.00 | 0.17 p=0.21 | -0.02 p=0.87 |
| Ca | | | | | | | | | | | | | | | **0.54** p=0.00 | **0.43** p=0.00 | **0.65** p=0.00 | **0.37** p=0.00 | -0.09 p=0.51 |
| Ca/Mg | | | | | | | | | | | | | | | | 0.20 p=0.14 | 0.13 p=0.31 | 0.17 p=0.19 | -0.04 p=0.76 |
| Fe | | | | | | | | | | | | | | | | | **0.32** p=0.02 | **0.32** p=0.02 | **-0.39** p=0.00 |
| Cu | | | | | | | | | | | | | | | | | | 0.16 p=0.23 | -0.00 p=0.99 |
| Zn | | | | | | | | | | | | | | | | | | | -0.03 p=0.82 |
| P | | | | | | | | | | | | | | | | | | | |

**Note:**
Values in bold indicate statistical significance.

*vs.* Fe (ρ = 0.35, *p* = 0.00). Ca levels were correlated with Fe and Cu (ρ = 0.42, *p* = 0.00; ρ = 0.62, *p* = 0.00 respectively), and Fe levels with Cu and P (ρ = 0.22, *p* = 0.01; ρ = −0.33, *p* = 0.00). The Ca/Mg ratio showed a weak positive correlation with Zn (ρ = 0.16, *p* = 0.05) (Table 11).

Correlation analysis of the variables in group 2 according to the NYHA classification revealed statistically significant negative relationships for BMI *vs.* smoking (ρ = −0.38, *p* = 0.00) and positive relationships for BMI *vs.* TG (ρ = 0.42, *p* = 0.02). Cigarette smoking was significantly positively correlated with Mg, Ca and Fe (respectively: ρ = 0.30, *p* = 0.01; ρ = 0.26, *p* = 0.03; ρ = 0.31, *p* = 0.01). For patients in this group who had diabetes, correlations were observed with hypertension, CRP, CH, LDL and Ca/Mg (respectively: ρ = 0.33, *p* = 0.00; ρ = 0.34, *p* = 0.00; ρ = −0.36, *p* = 0.03; ρ = −0.43, *p* = 0.02; ρ = 0.26, *p* = 0.03). Relatively moderate and strong dependencies were found for hypertension *vs.* CH and LDL (respectively: ρ = −0.49, *p* = 0.00 and ρ = −0.56, *p* = 0.00) as well as for myocardial infarction *vs.* TG and HDL (ρ = 0.41, *p* = 0.02 and ρ = −0.45, *p* = 0.01). In terms of the analysed biochemical parameters, CRP and creatinine showed significant correlations with urea (respectively: ρ = 0.30, *p* = 0.01; ρ = 0.63, *p* = 0.00). Statistically significant relationships were also noted for CH *vs.* creatinine and CH *vs.* urea (respectively: ρ = −0.34, *p* = 0.05; ρ = −0.36, *p* = 0.03). Notably, there were significant strong correlations for CH *vs.* LDL (ρ = 0.57, *p* = 0.00). Zn levels were negatively correlated with HDL (ρ = −0.49, *p* = 0.01), and relationships were also identified for Fe *vs.* Mg and Ca (respectively: ρ = 0.44, *p* = 0.00; ρ = 0.48, *p* = 0.00). In terms of the analysed minerals in group 2, statistically significant relationships were observed for Mg *vs.* Ca, Fe and Cu (respectively: ρ = 0.63, *p* = 0.00; ρ = 0.44, *p* = 0.00; ρ = 0.47, *p* = 0.00), Ca *vs.* Ca/Mg, Fe and Cu (respectively: ρ = 0.59, *p* = 0.00; ρ = 0.48, *p* = 0.00; ρ = 0.65, *p* = 0.00), and also between Fe and Zn (ρ = 0.43, *p* = 0.00). A positive correlation was shown between Ca/Mg *vs.* Cu (ρ = 0.30, *p* = 0.01) (Table 12).

## DISCUSSION

This is probably the first report exploring the relationships between elemental concentrations in serum for Polish patients with HF and anthropometric measurements as well as biochemical parameters. According to the literature data, the observed deficiencies in micronutrients and elements are common in outpatients with congestive heart failure (*Sattler et al., 2019*). They have also been conclusively linked to adverse clinical outcomes. In hospitalised HF patients, poor nutritional status is usually found in the elderly, whose blood parameters tend to show depleted micronutrient content, including vitamins: B1, B2, B6, C, D and E, folic acid, Ca, Mg, Fe, selenium (Se). Furthermore, significantly reduced levels of albumin, CH, TG, coenzyme Q10, creatinine and elevated CRP are observed (*Chery et al., 2019*; *Sciatti et al., 2016*; *McKeag et al., 2012*; *Lemon et al., 2010*; *Arcand et al., 2009*).

In our study, given the links between HF pathophysiology, clinical risk markers, anthropometric parameters and results of the patients' biochemical analysis, the majority of the studied patients presented elevated levels of CRP, CH, TG, LDL, creatinine, urea and depleted HDL. Moreover, deficient serum concentrations of Mg (47% of the studied

**Table 11 Correlation analysis of the studied parameters for group 1 (NYHA I and II).**

| | Smoking | Diabetes mellitus | Hyper­tension | MI | CRP | Creatinine | Urea | TG | CH | HDL | LDL | Mg | Ca | Ca/Mg | Fe | Cu | Zn | P |
|---|---|---|---|---|---|---|---|---|---|---|---|---|---|---|---|---|---|---|
| BMI | −0.07 p = 0.38 | **0.29** p = 0.00 | **0.29** p = 0.00 | 0.08 p = 0.35 | 0.03 p = 0.68 | 0.07 p = 0.38 | 0.01 p = 0.93 | **0.32** p = 0.00 | 0.09 p = 0.38 | **−0.46** p = 0.00 | 0.10 p = 0.39 | 0.07 p = 0.38 | **0.19** p = 0.02 | **0.17** p = 0.04 | =0.02 p = 0.80 | 0.11 p = 0.18 | 0.01 p = 0.93 | 0.09 p = 0.31 |
| Smoking | | **−0.18** p = 0.03 | −0.13 p = 0.13 | 0.11 p = 0.21 | 0.12 p = 0.15 | **−0.23** p = 0.00 | **−0.36** p = 0.00 | −0.13 p = 0.26 | −0.11 p = 0.32 | 0.16 p = 0.17 | −0.17 p = 0.13 | 0.08 p = 0.35 | 0.13 p = 0.13 | 0.08 p = 0.32 | −0.02 p = 0.84 | **0.21** p = 0.01 | −0.03 p = 0.68 | 0.06 p = 0.48 |
| Diabetes mellitus | | | **0.21** p = 0.01 | 0.07 p = 0.38 | −0.07 p = 0.38 | **0.28** p = 0.00 | **0.21** p = 0.01 | 0.08 p = 0.46 | 0.01 p = 0.91 | −0.22 p = 0.06 | 0.05 p = 0.64 | −0.09 p = 0.30 | 0.07 p = 0.43 | **0.22** p = 0.01 | −0.03 p = 0.68 | 0.02 p = 0.83 | 0.02 p = 0.81 | 0.01 p = 0.94 |
| Hypertension | | | | 0.06 p = 0.48 | −0.09 p = 0.31 | 0.07 p = 0.42 | 0.15 p = 0.08 | −0.07 p = 0.55 | −0.06 p = 0.60 | −0.11 p = 0.37 | −0.06 0.60 | 0.11 p = 0.20 | **0.24** p = 0.00 | **0.19** p = 0.02 | 0.13 p = 0.14 | 0.13 p = 0.12 | 0.05 p = 0.59 | −0.05 p = 0.59 |
| MI | | | | | −0.05 p = 0.54 | 0.12 p = 0.15 | 0.03 p = 0.76 | 0.03 p = 0.76 | 0.01 p = 0.89 | 0.02 p = 0.85 | 0.07 p = 0.52 | 0.03 p = 0.73 | 0.05 p = 0.53 | 0.03 0.71 | −0.07 p = 0.43 | −0.01 p = 0.94 | −0.02 p = 0.80 | −0.01 p = 0.86 |
| CRP | | | | | | −0.03 p = 0.73 | 0.08 p = 0.34 | 0.12 p = 0.30 | 0.01 p = 0.92 | 0.09 p = 0.43 | −0.01 p = 0.93 | 0.07 p = 0.37 | −0.04 p = 0.60 | **−0.18** p = 0.03 | **−0.21** p = 0.01 | **0.28** p = 0.00 | 0.01 p = 0.92 | −0.03 p = 0.75 |
| Creatinine | | | | | | | **0.53** p = 0.00 | −0.02 p = 0.84 | 0.08 p = 0.46 | 0.05 p = 0.64 | 0.06 p = 0.63 | −0.00 p = 0.99 | 0.06 p = 0.46 | 0.05 p = 0.56 | 0.04 p = 0.66 | 0.01 p = 0.95 | −0.08 p = 0.32 | **−0.20** p = 0.02 |
| Urea | | | | | | | | −0.21 p = 0.06 | 0.02 p = 0.82 | 0.13 p = 0.28 | 0.05 p = 0.68 | −0.03 p = 0.68 | −0.11 p = 0.19 | −0.07 p = 0.39 | −0.01 p = 0.90 | 0.04 p = 0.67 | −0.05 p = 0.59 | −0.10 p = 0.21 |
| TG | | | | | | | | | **0.54** p = 0.00 | −0.22 p = 0.06 | **0.46** p = 0.00 | 0.17 p = 0.13 | 0.18 p = 0.12 | 0.01 p = 0.96 | −0.00 p = 0.98 | **0.24** p = 0.03 | −0.11 p = 0.33 | 0.02 p = 0.89 |
| CH | | | | | | | | | | −0.10 p = 0.38 | **0.81** p = 0.00 | 0.11 p = 0.33 | 0.05 p = 0.64 | 0.04 p = 0.72 | −0.14 p = 0.20 | 0.21 p = 0.05 | −0.14 p = 0.19 | 0.06 p = 0.58 |
| HDL | | | | | | | | | | | −0.08 p = 0.52 | −0.20 p = 0.10 | **−0.30** p = 0.01 | **−0.24** p = 0.04 | −0.12 p = 0.31 | −0.19 p = 0.10 | −0.05 p = 0.64 | −0.11 p = 0.33 |
| LDL | | | | | | | | | | | | 0.10 p = 0.36 | 0.05 p = 0.68 | −0.01 p = 0.95 | −0.01 p = 0.95 | 0.08 p = 0.48 | −0.07 p = 0.57 | 0.05 p = 0.66 |
| Mg | | | | | | | | | | | | | **0.68** p = 0.00 | **−0.26** p = 0.00 | **0.35** p = 0.00 | **0.67** p = 0.00 | 0.08 p = 0.34 | −0.09 p = 0.29 |
| Ca | | | | | | | | | | | | | | **0.45** p = 0.00 | **0.42** p = 0.00 | **0.62** p = 0.00 | **0.22** p = 0.01 | −0.10 p = 0.21 |
| Ca/Mg | | | | | | | | | | | | | | | 0.14 p = 0.09 | 0.07 p = 0.43 | −0.05 p = 0.50 | −0.01 p = 0.92 |
| Fe | | | | | | | | | | | | | | | | **0.22** p = 0.01 | **0.16** p = 0.05 | **−0.33** p = 0.00 |
| Cu | | | | | | | | | | | | | | | | | −0.00 p = 0.99 | −0.05 p = 0.92 |
| Zn | | | | | | | | | | | | | | | | | | −0.01 p = 0.51 |
| P | | | | | | | | | | | | | | | | | | 0.09 p = 0.31 |

**Note:**
Values in bold indicate statistical significance.

**Table 12 Correlation analysis of the studied parameters for group 2 (NYHA III and IV).**

| | Smoking | Diabetes mellitus | Hyper-tension | MI | CRP | Creatinine | Urea | TG | CH | HDL | LDL | Mg | Ca | Ca/Mg | Fe | Cu | Zn | P |
|---|---|---|---|---|---|---|---|---|---|---|---|---|---|---|---|---|---|---|
| BMI | **−0.38** p = 0.00 | 0.21 p = 0.08 | −0.003 p = 0.98 | 0.20 p = 0.09 | 0.12 p = 0.33 | −0.07 p = 0.56 | −0.05 p = 0.68 | **0.42** p = 0.02 | −0.04 p = 0.81 | −0.15 p = 0.44 | −0.08 p = 0.67 | −0.12 p = 0.31 | −0.18 p = 0.14 | −0.02 p = 0.90 | −0.20 p = 0.10 | −0.16 p = 0.18 | 0.09 p = 0.45 | 0.02 p = 0.87 |
| Smoking | | −0.14 p = 0.24 | 0.08 p = 0.53 | −0.22 p = 0.06 | 0.03 p = 0.84 | −0.09 p = 0.48 | −0.01 p = 0.97 | −0.10 p = 0.59 | 0.07 p = 0.67 | 0.11 p = 0.55 | 0.08 p = 0.69 | **0.30** p = 0.01 | **0.26** p = 0.03 | 0.03 p = 0.78 | **0.31** p = 0.01 | 0.08 p = 0.53 | −0.02 p = 0.88 | 0.03 p = 0.78 |
| Diabetes mellitus | | | **0.33** p = 0.00 | 0.25 p = 0.03 | **−0.34** p = 0.00 | 0.04 p = 0.75 | 0.09 p = 0.47 | −0.04 p = 0.83 | **−0.36** p = 0.03 | 0.03 p = 0.88 | **−0.43** p = 0.02 | −0.13 p = 0.27 | 0.06 p = 0.62 | **0.26** p = 0.03 | −0.00 p = 0.97 | −0.23 p = 0.06 | −0.07 p = 0.56 | 0.06 p = 0.60 |
| Hypertension | | | | −0.02 p = 0.88 | −0.02 p = 0.85 | 0.14 p = 0.23 | 0.21 p = 0.07 | −0.26 p = 0.15 | **−0.49** p = 0.00 | −0.10 p = 0.58 | **−0.56** p = 0.00 | −0.08 p = 0.50 | 0.01 p = 0.93 | 0.12 p = 0.32 | −0.04 p = 0.76 | 0.02 p = 0.89 | 0.11 p = 0.37 | 0.10 p = 0.39 |
| MI | | | | | 0.00 p = 0.98 | −0.02 p = 0.86 | 0.01 p = 0.95 | **0.41** p = 0.02 | 0.04 p = 0.84 | **−0.45** p = 0.01 | −0.26 p = 0.17 | 0.02 p = 0.85 | 0.13 p = 0.27 | 0.16 p = 0.18 | 0.09 p = 0.44 | −0.17 p = 0.17 | 0.22 p = 0.07 | −0.17 p = 0.15 |
| CRP | | | | | | 0.23 p = 0.05 | **0.30** p = 0.01 | 0.08 p = 0.66 | −0.24 p = 0.17 | −0.29 p = 0.11 | −0.25 p = 0.18 | −0.06 p = 0.62 | −0.17 p = 0.15 | −0.11 p = 0.38 | −0.17 p = 0.17 | 0.22 p = 0.06 | −0.08 p = 0.49 | −0.11 p = 0.38 |
| Creatinine | | | | | | | **0.63** p = 0.00 | −0.21 p = 0.25 | −0.34 p = 0.05 | −0.21 p = 0.25 | −0.33 p = 0.08 | 0.02 p = 0.88 | −0.04 p = 0.72 | 0.03 p = 0.80 | 0.04 p = 0.75 | 0.19 p = 0.11 | −0.10 p = 0.43 | −0.14 p = 0.26 |
| Urea | | | | | | | | −0.14 p = 0.43 | **−0.36** p = 0.03 | −0.18 p = 0.34 | **−0.38** p = 0.04 | −0.00 p = 0.98 | −0.05 p = 0.71 | −0.01 p = 0.96 | −0.05 p = 0.67 | 0.20 p = 0.10 | −0.16 p = 0.20 | −0.13 p = 0.30 |
| TG | | | | | | | | | 0.16 p = 0.36 | −0.26 p = 0.16 | 0.01 p = 0.96 | −0.26 p = 0.15 | 0.02 p = 0.91 | 0.29 p = 0.10 | 0.07 p = 0.69 | −0.31 p = 0.08 | **0.36** p = 0.04 | −0.20 p = 0.27 |
| CH | | | | | | | | | | −0.28 p = 0.13 | **0.57** p = 0.00 | 0.15 p = 0.37 | 0.21 p = 0.22 | 0.15 p = 0.37 | 0.02 p = 0.91 | 0.11 p = 0.54 | −0.22 p = 0.20 | 0.28 p = 0.10 |
| HDL | | | | | | | | | | | 0.30 p = 0.11 | 0.05 p = 0.80 | −0.12 p = 0.50 | −0.08 p = 0.68 | −0.27 p = 0.15 | −0.10 p = 0.59 | **−0.49** p = 0.01 | 0.35 p = 0.05 |
| LDL | | | | | | | | | | | | 0.30 p = 0.11 | 0.15 p = 0.42 | −0.04 p = 0.82 | 0.05 p = 0.79 | −0.02 p = 0.91 | −0.10 p = 0.61 | 0.29 p = 0.12 |
| Mg | | | | | | | | | | | | | **0.63** p = 0.00 | −0.22 p = 0.07 | **0.44** p = 0.00 | **0.47** p = 0.00 | 0.07 p = 0.57 | **−0.24** p = 0.05 |
| Ca | | | | | | | | | | | | | | **0.59** p = 0.00 | **0.48** p = 0.00 | **0.65** p = 0.00 | 0.19 p = 0.11 | −0.20 p = 0.10 |
| Ca/Mg | | | | | | | | | | | | | | | 0.15 p = 0.22 | **0.30** p = 0.01 | 0.14 p = 0.25 | 0.06 p = 0.61 |
| Fe | | | | | | | | | | | | | | | | 0.16 p = 0.19 | **0.43** p = 0.00 | −0.23 p = 0.05 |
| Cu | | | | | | | | | | | | | | | | | −0.09 p = 0.44 | −0.15 p = 0.21 |
| Zn | | | | | | | | | | | | | | | | | | −0.05 p = 0.67 |
| P | | | | | | | | | | | | | | | | | | |

**Note:**
Values in bold indicate statistical significance.

men and 54% of the women) and Cu (in 44% of men and more than 31% of women) were observed, as well as subnormal serum Fe (2% of women) and Zn (1% of men). On the other hand, elevated Ca in serum was found in 50% of men and 49% of women (Refer to Table 5). In 44% of the studied men and 51% of the studied women, P levels in serum were also above-average. In our study, biochemical parameters were for the most part significantly correlated with each other, similarly to other studies (*Cosentino et al., 2019*; *Schröder et al., 2003*). The high percentage of overweight and obese participants observed in this study may be a reflection of poor eating habits and low physical activity, affecting equally men and women (Refer to Table 1). In our study, patients with HF in III–IV class of NYHA showed positive correlations between CH, LDL, TG levels and the presence of diabetes and arterial hypertension (Refer to Table 12), consistent with the literature reports (*Otsuka et al., 2016*). We have also demonstrated strong positive correlations in patients with HF in I–II class of NYHA for serum Ca *vs*. Mg, Cu *vs*. Mg and Cu *vs*. Ca (Refer to Table 11). Strong positive correlations in patients with HF in III-IV class of NYHA were found for Ca *vs*. Mg, Ca/Mg *vs*. Ca, Cu *vs*. Mg, Cu *vs*. Ca (Refer to Table 12), which is in agreement with reports from other authors (*Weber, Weglicki & Simpson, 2009*).

Interestingly, some 80% of the studied patients who were found to have macro-and micromineral deficiencies were also obese (with abnormally high BMI scores). A 2018 cohort study demonstrated that severe malnutrition due to elemental deficiencies was rare among obese patients with HF (*Sze et al., 2018*). Our findings also suggest increased inflammatory status (elevated CRP levels) and signs of metabolic syndrome (arterial hypertension, diabetes, obesity, dyslipidemia) in the studied group, which are risk factors for the development of ischemic heart disease (*Perrone-Filardi et al., 2015*). In the 2019 study by *Sikora et al. (2019)* among both the men and women studied, elemental concentrations which showed statistically significant correlations with tobacco smoking were also found to have a statistically significant influence on explaining their variation in regression analysis. Cigarette smoking was included in our analysis, too, but no significant strong correlations were noted between smoking and concentrations of the studied elements. Please note that smoking is also a sociological factor, closely linked to the place of residence, educational background and calendar age. The patients included in this study were already in the care of a cardiologist, and so they had made the decision to seek treatment/recovery and had, for the most part, quit smoking.

In the literature, it was demonstrated that alterations in the concentrations of both macro-and microminerals may impact on heart failure (*Sattler et al., 2019*; *Weber, Weglicki & Simpson, 2009*). Disturbed mineral homeostasis in serum, related to the development and progression of HF, may contribute to adverse structural remodelling of the heart and affect its function. Moreover, blood mineral concentrations reflect the patient's overall clinical status (*Demirbag et al., 2005*; *Takano et al., 2003*). In our study, we observed lower serum Mg levels, which were strongly positively correlated with increased Ca and reduced Cu levels in both groups 1 and 2 in the studied men and women. However, interactions in Ca homeostasis disorders are seldom isolated, usually being accompanied by increased concentrations of ionised Mg, a natural intracellular calcium
antagonist, and this was observed in our study, too. Hypomagnesemia was demonstrated to affect people with diabetes, metabolic syndrome, alcoholism, HIV, on medications affecting Mg absorption and in cancer patients (*Deheinzelin et al., 2000*). It has been reported that low Mg levels are found in people with myocardial infarction, and hereditary hypomagnesemia may lead to cardiomyopathy and heart failure. These disorders lead to myocardial necrosis, neuromuscular hyperexcitability, arrhythmia and increased oxidative stress (*Kramer, Phillips & Weglicki, 1997*; *Rubeiz et al., 1993*; *Riggs et al., 1992*). In turn, about 50% of the participants presented increased Ca concentrations, correlated with the levels of Mg, Ca/Mg, Fe, Zn in both groups 1 and 2 (Refer to Tables 11 and 12). In the literature, increases in calcium ions were reported in response to oxidative stress and in cardiomyocyte necrosis (*Weber, Weglicki & Simpson, 2009*). In heart failure, alterations in Fe levels, including depletion, are some of the most common coexisting symptoms, potentially affecting 37–61% of patients (*Jankowska et al., 2013*; *Enjuanes et al., 2016*). What is more, according to the 2016 guidelines of the European Society of Cardiology (ESC) on acute and chronic heart failure, HF patients should be checked for iron deficiency (*Ponikowski et al., 2016a*, *2016b*). In our study, low Fe was found only in 2% of the studied women. Blood Fe concentrations in the studied men and women were moderately positively correlated with those of Ca, and in men additionally with Mg (Refer to Tables 9 and 10). In patients with HF in I–II class of NYHA, Fe concentrations were moderately positively correlated with Mg and Ca (Refer to Table 11). In turn, in patients with HF in III-IV class of NYHA, Fe was moderately positively correlated with Ca and Mg (Refer to Table 12).

In the literature, there is evidence of a relationship between disruptions in trace element homeostasis, including Zn and Cu, and heart failure. It has also been demonstrated that the levels of Zn and Cu ions affect each other, and reduced Zn content is linked to impaired Cu metabolism (*Milne, Davis & Nielsen, 2001*). On the other hand, high copper intake may result in elevated Cr concentrations and reduced Mn concentrations in the heart (*Shen et al., 2005*). Our observations revealed positive correlations for Cu also with Mg and Ca, in both groups 1 and 2 of the studied patients (Refer to Tables 11 and 12), suggesting a mutual interaction between those elements in HF patients. There are sources reporting that blood Cu levels in HF patients may be higher than in the healthy population (*Singh et al., 1985*; *Málek et al., 2003*; *Huang et al., 2019*), but the phenomenon has not been explained to date. It has been suggested that elevated blood Cu levels are linked to increased levels of ceruloplasmin, a copper-binding protein (*Huang et al., 2019*). In our study, Cu levels were normal in the majority of the study group (Refer to Tables 5 and 6). However, while diabetes patients (who accounted for nearly 37% of our study group) (Refer to Table 2) may have the same serum Cu content as healthy individuals (*Cooper et al., 2005*), they also tend to have impaired copper metabolism, manifesting itself as myocardial copper deficiency coupled with defective cellular copper transport (*Zhang et al., 2014*). Diabetics have a pool of Cu(II) in the extracellular space, contributing to oxidative stress and mitochondrial dysfunction

(*Cooper et al., 2005*; *Lu et al., 2013*). This leads to copper-catalysed glycoxidation of the extracellular matrix, causing diabetic cardiomyopathy (*Cooper et al., 2004*; *Lu et al., 2013*). In our study, CRP was positively correlated with Cu, which may be a sign of the alterations taking place as a result of defective copper regulation.

It is also important to note the role of Zn ions, whose concentration impacts on endogenous antioxidant defence through metalloenzymes such as for instance superoxide dysmutase (SOD) (*Vasto et al., 2006*; *Yu et al., 2018*). In our study, zinc deficiency was found in 1% of men (Refer to Table 5). However, the literature data on the effects of changes in serum Zn levels in HF patients are inconsistent. Some studies have found decreased Zn concentrations in HF patients, but there have also been reports where no significant differences were observed in terms of serum Zn levels between HF patients and the control group (*Kosar et al., 2006*; *Ghaemian et al., 2011*; *Alexanian et al., 2014*; *Salehifar et al., 2008*; *Shokrzadeh et al., 2009*). Zn plays a major role in controlling cardiac contractility in cardiomyocytes, which may ultimately improve cardiac function (*Turan & Tuncay, 2017*), hence maintaining normal Zn levels seems to serve a protective role in HF patients.

The ICP-OES method used in this study is used to determine total serum phosphorus (organic and inorganic P) (*Shwiekh et al., 2013*), and that is why its concentrations observed in our study cannot be compared to the reference range of phosphate concentrations applicable to routine tests. In the study carried out by *Wach et al. (2018)*, the mean serum P in healthy patients fell in the range of 121–173 mg/L (*Wach et al., 2018*). This is significantly less than the P concentrations observed in our patients in both groups 1 and 2 (Refer to Table 8), but the serum P concentrations in our study were above-average in nearly half of the patients. It has been demonstrated that high serum P leads to arterial and valvular calcifications (*Disthabanchong, 2018*). Seeing as blood P content is regulated by the kidneys, hyperphosphatemia is often observed in patients with chronic renal failure (*Christopoulou et al., 2017*). In our study, in both groups 1 and 2, a small percentage of patients had elevated levels of creatinine–a measure of kidney function (Refer to Table 1 and 2). However, research findings suggest that vascular calcification begins while P levels are still within the normal range, pointing to healthy kidney function at the time (*Disthabanchong, 2018*). Increased P levels in serum also push up fibroblast growth factor 23 (FGF-23), leading to cardiac hypertrophy and adverse cardiovascular outcomes (*Ferrari, Bonjour & Rizzoli, 2005*; *Nishida et al., 2006*; *Burnett et al., 2006*). Enlargement of the heart is associated with impaired contractility, consequently leading to heart failure. Increased extracellular phosphate levels are also toxic to endothelial cells, promote CPP formation and induce VSMC transformation to an osteogenic phenotype (*Disthabanchong, 2018*).

Likewise, abnormally high levels of CRP, a marker of inflammation, observed in patients in our study, along with findings of diabetes in a large percentage of our patients, and dyslipidemia (low HDL, high TG, LDL), correlated with high BMI scores (Refer to Tables 1, 2, 11, 12), are factors conducive to atherosclerosis and development of heart disease (*Jaroszyński & Jaroszyńska, 2011*).

## CONCLUSIONS

Our study confirms the role of known risk factors in the development of heart failure, including: overweight, diabetes, hypertension, high triglycerides (TG), high total cholesterol (CH), high levels of low density protein (LDL) and reduced levels of high density protein (HDL), high CRP, high creatinine. Changes in the serum concentrations of macro-and microelements may significantly affect the severity of HF in patients with chronic heart failure.

The problem of chronic heart failure is a current medical issue affecting the worldwide population. The high mortality rates due to heart conditions inspire scholars to analyse the effects of many different risk factors for chronic heart failure. These factors include physiological, biochemical, anthropometric, as well as social and cultural parameters. Cardiac health is also affected by the patients' daily physical activity, as well as positive interpersonal relations (avoiding isolation), contact with nature, having a hobby, a positive attitude to life. The impact of those non-medical factors is still underestimated and difficult to measure, while the intensity of impact on heart health may be highly individual. We also wish to emphasise that differences in geographical location or ethnic origin may also modulate the influence of micro-and macroelements on the development of HF in populations *via* differences in lifestyle, diet and local risk factors contributing to HF. The antropometric and biochemical factors may also contribute to the development of chronic heart failure, seeing as the progression of circulatory failure is a complex process. Still, researchers emphasise that heart failure is already becoming a "cardiological epidemic" whose incidence is anticipated to rise continually. The knowledge on the subject-matter discussed in this paper still requires detailed, long-term and labour-intensive studies at various cardiological facilities.

## ACKNOWLEDGEMENTS

Our sincere thanks go out to the Patients who were involved in this study.

### Funding

The study was financed by internal funding of the Pomeranian Medical University, Szczecin, Poland. No other external funding was received. The funders had no role in study design, data collection and analysis, decision to publish, or preparation of the manuscript.

### Grant Disclosures

The following grant information was disclosed by the authors:
Pomeranian Medical University, Szczecin, Poland.

### Competing Interests

The authors declare that they have no competing interests.

## Author Contributions

- Iwona Gorący conceived and designed the experiments, performed the experiments, analyzed the data, authored or reviewed drafts of the paper, and approved the final draft.
- Ewa Rębacz-Maron analyzed the data, prepared figures and/or tables, authored or reviewed drafts of the paper, and approved the final draft.
- Jan Korbecki performed the experiments, authored or reviewed drafts of the paper, and approved the final draft.
- Jarosław Gorący performed the experiments, authored or reviewed drafts of the paper, and approved the final draft.

## Human Ethics

The following information was supplied relating to ethical approvals (*i.e.*, approving body and any reference numbers):

The study protocol was approved by the Ethics Committee of the Pomeranian Medical University in Szczecin (KB-0012/87/17 of 19.06.2017), subject to obtaining formal informed consent from all participants.

## Data Availability

The raw measurements are available in the Supplemental File.

## Supplemental Information

Supplemental information for this article can be found online at http://dx.doi.org/10.7717/peerj.12207#supplemental-information.

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
