# Peer review of "Concentrations of Mg, Ca, Fe, Cu, Zn, P and anthropometric and biochemical parameters in adults with chronic heart failure"

_PeerJ, doi:10.7717/peerj.12207_

## Round 0.1 · original submission · Minor Revisions

Authors, please kindly attend to comments raised by the reviewers. They have considered your work very favourably but raised minor concerns. Please, carefully attend to all comments raised, diligently. In addition, Editor encourages authors to address the following:

All script: Kindly indicate (Please place Table(s)/Figure(s) ?? here) in the specific places where you would like them to be:

1) Introduction: This section is quite ok, but requires more information. Before the objective statement, it is important to reiterate: a) Why is it important to study anthropometric and biochemical parameters in men and women? b) Where is the gap that is existing that makes this study relevant? c) What kind of study is required to fill this gap, and how should such kinds of study be performed to contribute the new knowledge? After you state the objective, it is important to state, what is the end goal expectation, which should connect to this established gap in literature.

2) Material and methods:
Start this section by creating a new subsection captioned "Schematic overview of experimental program'. This has to be supported by a flow diagram showing the steps followed, from participant enrolment, how they were grouped, study strategy, up to the various experimental analyses. The importance of this is to carry the reader along appropriately into this study.
Please, try to properly allocate the desirable subsections. An idea could be:
-Enrolment and inclusion criteria of study participants
-Ethics approval
-Grouping of study participants
-Study strategy,
and so on

3) Discussion: Please, kindly indicate (Refer to Table(s) ??) beside all the places where specific results in specific tables are being referred to. It is important that effort be made to ensure that all the Tables referred to in the results are covered.

4) Conclusions: Start the conclusions by reiterating what the gap in literature was , why this study was relevant, and to what extent this current study has attempted to fill this gap. Before stating what this study confirms, state those very key important findings, very succinctly. This will help guide readers to why these risk factors are very crucial. What are directions for future studies? Kindly brainstorm and make your recommendations. This is very important, so that readers will see how this study, as well as future studies directly links to the existing body of knowledge

This is a very important piece of work. You have done very well, and I look forward to your revised manuscript.

Thank you very much

·

Basic reporting

Manuscript Number: No #59941
Title: “Concentrations of Mg, Ca, Fe, Cu, Zn, P and anthropometric and biochemical parameters in men and women with chronic heart failure”
I appreciate the opportunity to review the manuscript.
Recommendation: The article is interesting and brings an important perspective on the subject into the available literature. The article discusses the very important problem of chronic heart failure. In my opinion, the study is of scientific relevance, it is interesting and should be published in PeerJ.
The heart failure is one of the biggest challenges facing health care systems in Poland and in the world, both in terms of morbidity, mortality, burden resulting from disability and inability to work, as well as the use of system resources and health services.
Over 1.2 million people in Poland suffer from heart failure (data for 2018), and the mortality rate among them has been very high for years. About 40% of patients die within 5 years of diagnosis. The incidence of heart failure has slightly decreased in the last 5 years and is now around 125,000. new cases every year.
Research methods (inductively coupled plasma optical spectrometry (ICP-OES) is appropriate, and the description of the method is detailed (sample preparation, digestion method, parameters of measurement, precision of the method used to measure the concentration of elements studied). Certified reference materials have been used for the validation of the methods.
Strong points of the paper: topicality of the subject, the social utility, especially because heart failure is one of the biggest challenges facing health care systems in Poland and in the world.
Comments on the paper: The Title reflects the content. The Abstract is well written.
Introduction: The introduction orients the reader to the problem of the heart failure.
Materials and Methods: The study was conducted using modern and well-chosen methods, which guarantee the reliability of the results obtained. Statistical methods were properly selected.
Results: Tables are clear and easy to follow.
Discussion: The paper is clear and succinct.
Conclusions: The conclusions drawn from the study are appropriate.
I have a few small corrections:
1) In the text of the manuscript I noticed a gaps between the tables, this should be corrected technically.
2) Poland should be in all affiliations, not Polska.
3) Please note that there is serum throughout the text

Experimental design

Materials and Methods: The study was conducted using modern and well-chosen methods, which guarantee the reliability of the results obtained. Statistical methods were properly selected.
Results: Tables are clear and easy to follow.
Discussion: The paper is clear and succinct.
Conclusions: The conclusions drawn from the study are appropriate.

Validity of the findings

No comment

Additional comments

I have a few small corrections:
1) In the text of the manuscript I noticed a gaps between the tables, this should be corrected technically.
2) Poland should be in all affiliations, not Polska.
3) Please note that there is serum throughout the text

·

Basic reporting

Authors may consider the possibility of replacing “men” and “women” with just the word “Adults” in order to shorten the length of the title and render it more understandable to the general readership
 The abstract of the manuscript is well-written in good English and conforms to the Peer J Journal standards
 I humbly suggest that, the researchers consider the possibility of including a brief discussing of the neurohormonal basis of the pathophysiology of chronic heart failure. This approach will enable most readers to better understand the relevance of the study objectives and the relationship between the risk factors under investigation and heart failure as a chronic disease
 Although the researchers have attempted to discuss the relationship between the individual risk factors under investigation and chronic heart failure, the underlying physiochemical mechanisms that explains and establish these relationships remain unclear (For example: how will hypo or hypercalcemia affect prognosis in patients with chronic heart failure). These mechanisms should be briefly but explicitly described with proper references.
 Chronic heart failure classifications have been adequately discussed. However, there is no mention on how these patients were diagnosed with heart failure and whether diagnosis was conducted and or confirmed by a residency trained and board-certified cardiologist. This approach will give the general readership the opportunity to assess the validity of such diagnosis
 Generally, the language is clear and unambiguous, literature review was thorough and extensive and properly referenced.
 Figures are well labelled, very relevant and accurately described.
 The structure of the entire manuscript conforms to PeerJ Standards

Experimental design

The research objectives are clearly defined and concordant with the scope of the Journal
 The researchers have clearly stated and discussed how their results will definitely fill a knowledge gap within the scientific context
 The research question is also well defined, relevant but not that all meaningful
 The authors exercised due diligence in their investigative procedures and have exhibited some level of technical and ethical standards
 The use of only signs and symptoms to diagnose heart failure is not clinically prudent and constitute a scientifically unacceptable approach. The use of signs and symptoms plus electrocardiographic findings for the diagnosis of heart failure will be more appropriate.
 The researchers have also clearly indicated in their manuscripts that, study participants were divided into two groups. However, there is no mention of how these participants were selected and how they were assigned to each group. This approach has the great potential to create various sources of selection and allocation bias
 The conclusions are based on the study results. However, since this study can somehow be considered as a “me too” study, the findings more or less constitute a confirmation of existing knowledge on the topic
 Finally, the last sentence in the conclusion section “Changes in the serum concentrations of macro- and micronutrients may significantly affect the severity of HF in patients with chronic heart Failure “needs a little bit more of clarity
 Humble suggestion for consideration: “Abnormalities in the serum concentrations of macro-and micronutrients may significantly affect the prognosis in patients with chronic heart failure “.

Validity of the findings

Although the findings are can be considered relatively valid and relevant and very well supported by literature with a great potential to significantly expand the scope of the current literature on the topic, these findings do not meaningfully add any new knowledge to the already available knowledge on the topic. This is because the current topic under review has been exhaustedly studied and many findings reported in the literature as evidenced by the quality and quantity of citations used under the references
 All underlying data have been sufficiently made available and statistical methods and approaches are considered valid and appropriate based on the nature and type of data collected

Additional comments

No comments

---

## Round 0.2 · Minor Revisions

Thank you authors for revising your work based on reviewers' comments.

I am disappointed that the authors appeared to ignore the detailed comments provided to them, and addressed only the comments of the reviewers. It is not kind. It is an unacceptable practice.

The editor requests authors to kindly and diligently address them one by one, as copied and pasted below, which now has some additions to it:

1) Introduction: This section is quite ok, but requires more information. Before the objective statement, it is important to reiterate: a) Why is it important to study anthropometric and biochemical parameters in men and women? b) Where is the gap that is existing that makes this study relevant? c) What kind of study is required to fill this gap, and how should such kinds of study be performed to contribute the new knowledge? After you state the objective, it is important to state, what is the end goal expectation, which should connect to this established gap in the literature.

2) Material and methods:
Start this section by creating a new subsection captioned "Schematic overview of experimental program'. This has to be supported by a flow diagram showing the steps followed, from participant enrolment, how they were grouped, study strategy, up to the various experimental analyses. The importance of this is to carry the reader along appropriately into this study.
Please, try to properly allocate the desirable subsections.
This should follow this pattern
-Enrolment and inclusion criteria of study participants
-Ethics approval
-Grouping of study participants
-Study strategy,
and so on

3) Results are ok, however, kindly change spearman rank correlation coefficient to 'ρ' (rho), delete all 'Rs' (if you want to use it, you must write it properly, 'r' with superscript 's'. Therefore, kindly use 'ρ' (rho) sign.

In Tables 9, 10, 11, and 12, it is assumed that these values are correlation (ρ) values. Please, for each correlation value in the matrix, kindly insert below each corresponding p-values, so that readers can clearly know why those in bold are statistically significant.

In the results text where all correlation values are stated, kindly indicate each p-value (which must correspond to p-value in the matrix ofcourse)

When you say ' statistically significant weak relationship' what do you mean? There is nothing like a statistically significant weak relationship. It is either statistically significant or not. Period. That is why the p-values must be indicated in each one. p-values must be inserted in all correlation matrices.

How can you assert strong correlation when rho is 0.54 and 0.46 (Lines 398-399)??

The entire correlation section must be revised. Firstly, in your statistical analysis section, indicate a referenced criteria that you want to use to define strong, moderate and weak correlation. From this, to this, as either strong, moderate, or weak. Readers will know that this is the criteria author have set for all correlation outcomes, which must be based on a published reference.
In relaying the correlation outcomes/results, this criteria must be used, strictly with its terms 'strong', 'moderate', 'weak', whether it is positive or negative.

Where ' statistically significant' is used, p-values must be indicated. Therefore, all correlation Tables 9, 10, 11, and 12 must have p-values under each correlation coefficient.

3) Discussion: Please, kindly indicate (Refer to Table(s) ??) beside all the places where results of specific tables are being referred to. It is important that effort be made to ensure that all the Tables referred to in the results are covered.

4) Conclusions: Start the conclusions by reiterating what the gap in literature was , why this study was relevant, and to what extent this current study has attempted to fill this gap. Before stating what this study confirms, state those very key important findings, very succinctly. This will help guide readers to why these risk factors are very crucial. What are directions for future studies? Kindly brainstorm and make your recommendations. This is very important, so that readers will see how this study, as well as future studies directly links to the existing body of knowledge


Looking forward to your revised manuscript.

Thank you.

---

## Round 0.3 · accepted · Accept

I am very satisfied with the revised manuscript, and it can now be accepted for publication. The authors have benefited greatly from the peer-review process, which improved the quality of this work. Thank you authors for finding PeerJ as your journal of choice, and looking forward to your future scholarly contributions.

Congratulations and very best regards.